# Expert-Supervised Reinforcement Learning for Offline Policy Learning and Evaluation

**Aaron Sonabend-W**
Harvard University
asonabend@g.harvard.edu

**Junwei Lu**
Harvard University
junweilu@hsph.harvard.edu

**Leo A. Celi**
MIT
lceli@mit.edu

**Tianxi Cai**
Harvard University
tcai@hsph.harvard.edu

**Peter Szolovits**
MIT
psz@mit.edu

## Abstract

Offline Reinforcement Learning (RL) is a promising approach for learning optimal policies in environments where direct exploration is expensive or unfeasible. However, the adoption of such policies in practice is often challenging, as they are hard to interpret within the application context, and lack measures of uncertainty for the learned policy value and its decisions. To overcome these issues, we propose an Expert-Supervised RL (ESRL) framework which uses uncertainty quantification for offline policy learning. In particular, we have three contributions: 1) the method can learn safe and optimal policies through hypothesis testing, 2) ESRL allows for different levels of risk averse implementations tailored to the application context, and finally, 3) we propose a way to interpret ESRL's policy at every state through posterior distributions, and use this framework to compute off-policy value function posteriors. We provide theoretical guarantees for our estimators and regret bounds consistent with Posterior Sampling for RL (PSRL). Sample efficiency of ESRL is independent of the chosen risk aversion threshold and quality of the behavior policy.

## 1 Introduction

With increasing success in reinforcement learning (RL), there is broad interest in applying these methods to real-world settings. This has brought exciting progress in offline RL and off-policy policy evaluation (OPPE). These methods allow one to leverage observed data sets collected by expert exploration of environments where, due to costs or ethical reasons, direct exploration is not feasible. Sample-efficiency, reliability, and ease of interpretation are characteristics that offline RL methods must have in order to be used for real-world applications with high risks, where a tendency is exhibited towards sampling bias. In particular there is a need for policies that shed light into the decision-making at all states and actions, and account for the uncertainty inherent in the environment and in the data collection process. In healthcare data for example, there is a common bias that arises: drugs are mostly prescribed only to sick patients; and so naive methods can lead agents to consider them harmful. Actions need to be limited to policies which are similar to the expert behavior and sample size should be taken into account for decision-making [1, 2].

To address these deficits we propose an Expert-Supervised RL (ESRL) approach for offline learning based on Bayesian RL. This method yields safe and optimal policies as it learns when to adopt the expert's behavior and when to pursue alternative actions. Risk aversion might vary across applications as errors may entail a greater cost to human life or health, leading to variation in tolerance for the target policy to deviate from expert behavior. ESRL can accommodate different risk aversion levels.

We provide theoretical guarantees in the form of a regret bound for ESRL, independent of the risk aversion level. Finally, we propose a way to interpret ESRL's policy at every state through posterior distributions, and use this framework to compute off-policy value function posteriors for any given policy.

While training a policy, ESRL considers the reliability of the observed data to assess whether there is substantial benefit and certainty in deviating from the behavior policy, an important task in a context of limited data. This is embedded in the method by learning a policy that chooses between the optimal action or the behavior policy based on statistical hypothesis testing. The posteriors are used to test the hypothesis that the seemingly optimal action is indeed better than the one from the behavior policy. Therefore, ESRL is robust to the quality of the behavior policy used to generate the data.

To understand the intuition for why hypothesis testing works for offline policy learning, we discuss an example. Consider a medical setting where we are interested in the best policy to treat a complex disease over time. We first assume there is a standardized treatment guideline that works well and that most physicians adopt it to treat their patients. The observed data will have very little exploration of the whole environment —in this case, meaning little use of alternative treatments. However, the state-action pairs observed will be near optimal. For any fixed state, those actions not recommended by the treatment guidelines will be rare in the data set and the posterior distributions will be dominated by the uninformative wide priors. The posteriors for the value associated with the optimal actions will incorporate more information from the data as they are commonly observed. Thus, testing for the null hypothesis that an alternative action is better than the treatment guideline will likely yield a failure to reject the null, and the agent will conclude the physician's action is best. Unless the alternative is substantially better for a given state, the learned policy will not deviate from the expert's behavior when there is a clear standard of care.

On the other hand, if there is no treatment guideline or consensus among physicians, different doctors will try different strategies and state-action pairs will be more uniformly observed in the data. At any fixed state, some relatively good actions may have narrower posterior distributions associated with their value. Testing for the null hypothesis that a fixed action is better than what the majority of physicians chose is more likely to reject the null and point towards an alternative action in this case, as variance will be smaller across the sampled actions. Deviation from the (noisy) behavior policy will occur more frequently. Therefore, whether there is a clear care guideline or not, the method will have learned a suitable policy. A central point in Bayesian RL is that the posterior provides not just the expected value for each action, but also higher moments. We leverage this to produce interpretable policies which can be understood and analyzed within the context of the application. We illustrate this with posterior distributions and credible intervals (CI). We further propose a way to produce posterior distributions for OPPE with consistent and unbiased estimates.

**Handling Uncertainty.** To the best of our knowledge, there is no work that has incorporated hypothesis testing directly into the policy training process. However, accounting for the uncertainty in policy estimation is a successful idea which has been widely explored in other works. Methods range from confidence interval estimation using bootstrap, to model ensembles for guiding online exploration [3, 4, 5]. For example, a simple and effective way of incorporating uncertainty is through random ensembles (REM) [6]. These have shown promise on Atari games, significantly outperforming Deep $Q$ networks (DQN) [7] and naive ensemble methods in the offline setting. We adopt the Bayesian framework, which has been proven successful in online RL [8, 9], as it provides a natural way to formalize uncertainty in finite samples. Bayesian model free methods such as temporal difference (TD) learning provide provably efficient ways to explore the dynamics of the MDP [10, 11, 12]. Gaussian Process TD can be also used to provide posterior distributions with mean value and CI for every state-action pair [13]. Although efficient for online exploration, TD methods require large data in high dimensional settings, which can be a challenge in complex offline applications such as healthcare. ESRL is model-based which makes it sample efficient [14]. Within model-based methods, the Bayesian framework allows for natural incorporation of uncertainty measures. Posterior sampling RL proposed by Strens efficiently explores the environment by using a single MDP sample per episode [15]. ESRL fits within this line of methods, which are theoretically guaranteed to be efficient in terms of finite time regret bounds [16, 17].

**Hypothesis Testing for Offline RL** Naively applying model-based RL to offline, high dimensional tasks can degrade its performance, as the agent can be led to unexplored states where it fails to learn

reliable policies. There are environments where simple approaches like behavior cloning (BC) on the offline data set is enough to ensure reliability. BC has actually been shown to perform quite well in offline benchmarks like RL Unplugged [18], D4RL [19] and Atari when the data is collected from a single noisy behavior policy [20]. The issue with these approaches is that there is to be gained in terms of optimality with respect to the expert, and there is no guarantee that the learned policies are safe in all states, a necessary condition when treating patients. A common strategy is to regularize the learned policy towards the behavior policy whether directly in the state space or in the action space [18, 20, 21, 22, 23]. However, there are cases where the data logging policy is a noisy representation of the expert behavior, and regularization will lead to sub-optimal actions. ESRL can detect these cases through hypothesis testing [24] to check whether improvement upon the behavior policy is feasible and, if so, incorporate new actions into the policy in accordance with the user's risk tolerance. Additionally, as opposed to the regularization hyper-parameter that one must choose for methods like Batch Constrained deep Q-learning (BCQ) [18, 20], the risk-aversion parameter has a direct interpretation as the significance level that the user is comfortable with for the policy to deviate from the expert behavior. It allows the method to be tailored to different scientific and business applications where one might have different tolerance towards risk in search for higher rewards.

**Off-Policy Policy Evaluation and Interpretation.** Many of the aforementioned methods can be easily adapted for offline learning and often importance sampling is used to address the distribution shift between the behavior and target policies [25]. However, importance sampling can yield high variance estimates in finite samples, especially in long episodes. Doubly robust estimation of the value function is proposed to address these issues. These methods will have low variance and consistent estimators if either the behavior policy or the model is correctly specified [26, 27]. Still, in finite samples or environments with high dimensional state-action spaces, these doubly robust estimators may still not be reliable, because only a few episodes end up contributing to the actual value estimate due to the product in the importance sampling weights [2]. Additionally, having point estimates without any measure of associated uncertainty can be dangerous, as it is hard to know whether the sample size is large enough for the estimate to be reliable. To this end, we use the ESRL framework to sample MDP models from the posterior and evaluate the policy value. Our estimates are unbiased and consistent, and are equipped with uncertainty measures.

## 2 Problem Set-up

We are interested in learning policies that can be used in real-world applications. To develop the framework we will use the clinical example discussed in Section 1. Consider a finite horizon MDP defined by the following tuple: $< \mathcal{S}, \mathcal{A}, R^M, P^M, P_0, \tau >$, where $\mathcal{S}$ is the state-space, $\mathcal{A}$ is the action space, $M$ is the model over all rewards and state transition probabilities with prior $f(\cdot)$, $R^M(s,a) : \mathcal{S} \times \mathcal{A} \to [0,1]$ is the reward distribution for fixed state-action pair $(s,a)$ under model $M$, with mean $\bar{R}^M(s,a)$. $P_a^M(s'|s)$ is the probability distribution function for transitioning to state $s'$ from state-action pair $(s,a)$ under model $M$, $\tau \in \mathbb{N}$ is the fixed episode length, and $P_0$ is the initial state distribution. The true MDP model $M^*$ has distribution $f$.

The behavior policy function is a noisy version of a deterministic policy. Going back to the clinical example there is generally a consensus of what the correct treatment is for a disease, but the data will be generated by different physicians who might adhere to the consensus to varying degrees. Thus, we model the standard of care as a deterministic policy function $\pi^0 : \mathcal{S} \times \{1, \ldots, \tau\} \mapsto \mathcal{A}$. The behavior policy is $\pi(s,t) = \pi^0(s,t)$ with probability (w.p.) $1 - \epsilon$, and $\pi(s,t) = a$ sampled uniformly at random from $\mathcal{A}$ w.p. $\epsilon$. For a fixed $\epsilon \in [0,1]$, $\pi$ generates the observed data $\boldsymbol{D}_T = \{(s_{i1}, a_{i1}, r_{i1}, \ldots, s_{i\tau}, a_{i\tau}, r_{i\tau})\}_{i=1}^T$ which consists of $T$ episodes (i.e. patient treatment histories), where $s_{i1} \sim P_0 \; \forall i = 1, \ldots, T$. Note that $\pi^0$ may generally yield high rewards, however it is not necessarily optimal and can be improved upon.

We'll denote a policy function by $\mu : \mathcal{S} \times \{1, \ldots, \tau\} \to \mathcal{A}$. The associated value function for $\mu$, model $M$ is $V_{\mu,t}^M(s) = \mathbb{E}_{M,\mu}\left[\sum_{j=t}^\tau \bar{R}^M(s_j, a_j)|s_t = s\right]$, and the action-value function is $Q_{\mu,t}^M(s,a) = \bar{R}^M(s,a) + \sum_{s' \in \mathcal{S}} P_a^M(s'|s)V_{\mu,t+1}(s')$. At any fixed $(s,t)$, $\mu(s,t) \equiv \arg\max_a Q_{\tilde{\mu},t}(s,a)$, note that we allow $\tilde{\mu}$ in the $Q$ function to differ from $\mu$. This distinction will be useful as $\tilde{\mu}$ can be $\mu$, $\pi$ (or the ESRL policy defined in Section 3). Finally, $\pi(a|s,t)$ is the probability of $a$ given $(s,t)$, under the behavior policy.

# 3   Expert-Supervised Reinforcement Learning

We are interested in finding a policy which improves upon $\pi$. Directly regularizing the target policy to the behavior might restrict the agent from finding optimal actions, especially when $\pi$ has a high random component $\epsilon$, or $\pi^0$ is not close to optimal. Thus we want to know when to use $\mu$ versus $\pi$. This motivates the use of posterior distributions to quantify how well each state has been explored in $\boldsymbol{D}_T$ and how close $\pi$ is to $\pi^0$. At every state and time $(s,t)$ in the episode we can sample $K$ MDP models from $f(\cdot|\boldsymbol{D}_T)$. These samples are used to compare the quality of the behavior and target policy actions. We consider both the expected values of each action $Q_{\tilde{\mu},t}(s, \pi(s,t))$ versus $Q_{\tilde{\mu},t}(s, \mu(s,t))$, and their second moments for any fixed $\tilde{\mu}$. In particular, posterior distributions of $Q_{\tilde{\mu},t}(s,a)$, $a \in \mathcal{A}$ are used to test if the value for $\mu(s,t)$ is significantly better than $\pi$. This makes the learning process robust to the quality of the behavior policy. Next we formalize these arguments by a sampling scheme, define the ESRL policy, and state its theoretical properties.

**Sampling $Q$ functions.**   The distribution over the MDP model $f(\cdot|\boldsymbol{D}_T)$ implicitly defines a posterior distribution for any $Q$ function: $Q_{\tilde{\mu},t}(s,a) \sim f_Q(\cdot|s,a,t,\boldsymbol{D}_T)$. As the true MDP model $M^*$ is stochastic, we want to approximate the conditional mean $Q$ value: $\mathbb{E}\left[Q_{\tilde{\mu},t}^{M^*}(s,a)|s,a,t,\boldsymbol{D}_T\right]$. We do this by sampling $K$ MDP models $M_k$, compute $Q_{\tilde{\mu},t}^{(k)}(s,a)$, $k = 1,\ldots,K$ and use $\hat{Q}_{\tilde{\mu},t}(s,a) \equiv \frac{1}{K}\sum_{k=1}^{K}Q_{\tilde{\mu},t}^{(k)}(s,a)$.

**Lemma 3.1.** $\hat{Q}_{\tilde{\mu},t}(s,a)$ *is consistent and unbiased for* $Q_{\tilde{\mu},t}^{M^*}(s,a)$ :

$$\mathbb{E}\left[\hat{Q}_{\tilde{\mu},t}(s,a)|s,a,t,\boldsymbol{D}_T\right] = \mathbb{E}\left[Q_{\tilde{\mu},t}^{M^*}(s,a)|s,a,t,\boldsymbol{D}_T\right],$$

$$\hat{Q}_{\tilde{\mu},t}(s,a) - \mathbb{E}\left[Q_{\tilde{\mu},t}^{M^*}(s,a)|s,a,t,\boldsymbol{D}_T\right] = O_p\left(K^{-\frac{1}{2}}\right), \forall(t,s,a).$$

Lemma 3.1 establishes desirable properties for our $Q$ function estimation. Choosing $K = 1$ yields an immediate result: every $Q_{\tilde{\mu},t}^{(k)}(s,a)$ from model $M_k$ is unbiased.

The stochasticity of $M^*$ and $\pi$ suggests the mean $Q$ values for $\pi$ and $\mu$ are not enough to make a decision for whether it is beneficial to deviate from $\pi$. Next we discuss how to directly incorporate this uncertainty assessment into the policy training through Bayesian hypothesis testing.

**ESRL Policy Learning Through Hypothesis Testing.**   For a fixed $\alpha$-level, denote the ESRL policy by $\mu^\alpha$, we next describe the steps to learn this policy. By iterating backwards as in dynamic programming, assume we know $\mu^\alpha(s,j)$ $\forall s \in \mathcal{S}, j \in \{t+1,\ldots,\tau\}$, and we have $V_{\mu^\alpha,\tau+1}^M(s) = 0, \forall s \in \mathcal{S}$. Intuitively, at any $(s,t)$ we want to assess whether there is enough information in $D_T$ to support choosing the seemingly best action $\mu$ over $\pi$. Denote $\mu(s,t) = \arg\max_a Q_{\mu^\alpha,t}(s,a)$ as the best action if we follow the ESRL policy $\mu^\alpha$ onward, we formalize this with the following hypothesis:

$$H_0 : Q_{\mu^\alpha,t}^M(s,\mu(s,t)) \leq Q_{\mu^\alpha,t}^M(s,\pi(s,t)). \tag{1}$$

Note that in (1), both $Q$ functions assume the agent proceeds with ESRL policy $\mu^\alpha$ onward. If we can reject $H_0$, then it is safe to follow $\mu$, if we fail to reject the null, it does not necessarily mean the behavior policy is better, but there is not enough information in the data to support following $\mu$. To construct a safe ESRL policy we simply evaluate $H_0$ by computing the null probability $\mathbb{P}\left(H_0|t,s,\boldsymbol{D}_T\right)$, if this is below a pre-specified risk-aversion level $\alpha$ then we can safely choose $\mu$. In other words if the learned policy does not yield a significantly better value estimate, then we fail to reject the null and proceed to use the behavior policy's action. The ESRL policy at $(s,t)$ is then

$$\mu^\alpha(s,t) = \begin{cases} \mu(s,t) & \text{if} \quad \mathbb{P}\left(H_0|t,s,\boldsymbol{D}_T\right) < \alpha, \\ \pi(s,t) & \text{else.} \end{cases}$$

To compute $\mu^\alpha(s,t)$, we start by sampling $K$ MDP models from the posterior distribution, computing $\{Q_{\mu^\alpha,t}^{(k)}(s,a)\}_{k=1}^{K}$ and splitting the samples into two disjoint sets $\mathcal{I}_1, \mathcal{I}_2$. We use $\mathcal{I}_1$ to draw the policy $\hat{\mu}(s,t)$ based on majority voting. Then we use $\mathcal{I}_2$ to assess the null hypothesis in (1), with estimator $\hat{\mathbb{P}}\left(H_0|t,s,\boldsymbol{D}_T\right) = \frac{1}{K}\sum_{k=1}^{K} I\left(Q_{\mu^\alpha,t}^{(k)}(s,\hat{\mu}(s,t)) \leq Q_{\mu^\alpha,t}^{(k)}(s,\pi(s,t))\right)$. We next discuss convergence of the null probability estimator, and how to choose $\hat{\mu}(s,t)$ $\forall(s,t) \in \mathcal{S} \times \{1,\ldots,\tau\}$.

**Lemma 3.2.** *Let* $\mathbb{P}^*\left(H_0|t,s,\boldsymbol{D}_T\right)$ *be the null probability under true MDP* $M^*$ *with policy* $\mu^*$,

$$\hat{\mathbb{P}}\left(H_0|t,s,\boldsymbol{D}_T\right) - \mathbb{P}^*\left(H_0|t,s,\boldsymbol{D}_T\right) = O_p\left(K^{-\frac{1}{2}}\right).$$

Lemma 3.2 guarantees that we can construct a consistent policy $\mu^\alpha$ by sampling from the MDP posterior. There are two factors that come into play in (1): the difference in mean $Q$ values, and the second moments. If $Q_{\mu^\alpha,t}^{M^*}(s,\mu(s,t))$ is much higher than $Q_{\mu^\alpha,t}^{M^*}(s,\pi(s,t))$, but there are very few samples in $\boldsymbol{D}_T$ for $(s,\mu(s,t))$, the wide posterior will translate into a high $\hat{\mathbb{P}}\left(H_0|t,s,\boldsymbol{D}_T\right)$ leading ESRL to adopt $\pi(s,t)$. To choose $\mu(s,t)$ there needs to be both a substantial benefit for this new action and a high certainty of such gain. How averse the user is to deviating from $\pi$ is controlled by parameter $\alpha$. A small risk averse $\alpha$ will allow $\mu^\alpha$ to deviate from $\pi$ only with high certainty. When $\alpha = 1$, Algorithm 1 boils down to an offline version of PSRL after $T$ episodes, which uses majority voting for a robust policy. Algorithm 1 collects these ideas in order to learn an ESRL policy $\mu^\alpha$. Disjoint sets $\mathcal{I}_1, \mathcal{I}_2$, ensure independence and keep theoretical guarantees under the Assumption 3.3.

---

**Algorithm 1:** Expert-Supervised RL

---

Sample $M_k \sim f(\cdot|\boldsymbol{D}_T)$ $k = 1,\ldots,K$, set $\mathcal{I}_1 = \{1,\ldots,\lceil\frac{K}{2}\rceil\}$, $\mathcal{I}_2 = \{\lceil\frac{K}{2}\rceil + 1,\ldots,K\}$;

Set $\hat{V}_{\tau+1}^{(k)}(s) \leftarrow 0 \ \forall s \in \mathcal{S}, k = 1,\ldots,K$;

Compute behavior distribution $\pi(a|s,t)$ from $\boldsymbol{D}_T$, set $\pi(s,t) = \arg\max_a \pi(a|s,t)$;

**for** $t = \tau,\ldots,1$ **do**

    **for** $s \in \mathcal{S}$ **do**

        **for** $k = 1,\ldots,K$ **do**

            $\mu_k(s,t) \leftarrow \arg\max_a Q_{\mu^\alpha,t}^{(k)}(s,a)$;

        **end**

        $\hat{\mu}(s,t) \leftarrow$ maj. vote$\{\mu_k(s,t), k \in \mathcal{I}_1\}$;

        Compute $\hat{\mathbb{P}}(H_0|s,t,\boldsymbol{D}_T) = \frac{1}{|\mathcal{I}_2|}\sum_{k\in\mathcal{I}_2} I\left(Q_{\mu^\alpha,t}^{(k)}(s,\hat{\mu}(s,t)) < Q_{\mu^\alpha,t}^{(k)}(s,\pi(s,t))\right)$;

        **for** $k = 1,\ldots,K$ **do**

            $\mu_k^\alpha(s,t) \leftarrow I\left(\hat{\mathbb{P}}(H_0|s,t,\boldsymbol{D}_T) < \alpha\right)\mu_k(s,t) + I\left(\hat{\mathbb{P}}(H_0|s,t,\boldsymbol{D}_T) \geq \alpha\right)\pi(s,t)$;

            $\hat{V}_t^{(k)}(s) \leftarrow Q_{\mu^\alpha,t}^{(k)}(s,\mu_k^\alpha(s,t))$;

        **end**

        $\hat{\mu}^\alpha(s,t) \leftarrow$ maj. vote$\{\mu_k^\alpha(s,t), k \in \mathcal{I}_1\}$;

        $\mathcal{M}^\alpha(s,t) \leftarrow \{k | k \in \mathcal{I}_1, \mu_k^\alpha(s,t) = \hat{\mu}^\alpha(s,t)\}$;

    **end**

**end**

Define majority voting set: $\text{MV}^\alpha = \cap_{(s,t)}\mathcal{M}^\alpha(s,t)$;

**if** $\exists k \in MV^\alpha$ **then**

    choose $k \in \text{MV}^\alpha$ at random, set $k^{\text{MV}} \leftarrow k$

**else**

    Set $k^{\text{MV}}$ to most common $k \in \mathcal{M}^\alpha(s,t), \forall(s,t)$

**end**

Set $\mu^\alpha = \mu_{k^{MV}}$

---

**Assumption 3.3.** *Let* $\mathbb{P}^*(H_0|s,t,\boldsymbol{D}_T)$ *be defined as in* (1) *for the true* $M^*$. *The chosen risk-averse parameter* $\alpha \in [0,1]$ *satisfies* $\mathbb{P}^*(H_0|s,t,\boldsymbol{D}_T) \neq \alpha \ \forall(s,t) \in \mathcal{S} \times \{1,\ldots,\tau\}$.

As $\alpha$ is set by the user, Assumption 3.3 is easily satisfied as long as $\alpha$ is chosen carefully. Let $V_{\mu^{\alpha*},1}^{M^*}(s)$ be the value under the true MDP $M^*$ and let $\mu^{\alpha*}$ be an ESRL policy which uses the null hypotheses in (1) defined under $M^*$. Then, for episode $i$ we can define the regret for $\mu^\alpha$ from Algorithm 1 as $\Delta_i = \sum_{s_i\in\mathcal{S}} P_0(s_i)\left(V_{\mu^{\alpha*},1}^{M^*}(s_i) - V_{\mu^\alpha,1}^{M^*}(s_i)\right)$, and the expected regret after $T$ episodes as $\mathbb{E}[Regret(T)] = \mathbb{E}\left[\sum_{i=1}^T \Delta_i\right]$.

**Theorem 3.4** (Regret Bound for ESRL)**.** *For any* $\alpha \in [0,1]$ *which satisfies Assumption 3.3, Algorithm 1 using* $\boldsymbol{D}_T$ *and choosing* $K = \mathcal{O}(T)$ *will yield*

$$\mathbb{E}\left[Regret\left(T\right)\right] = \mathcal{O}\left(\tau S\sqrt{AT\log(SAT)}\right).$$

Theorem 3.4 shows ESRL is sample efficient, flexible to risk aversion level $\alpha$, and robust to the quality of behavior policy $\pi$. As the regret bound is true for any level of risk aversion $\alpha$, Algorithm 1 universally converges to the oracle. This makes ESRL flexible for a wide range of applications. It also shows that ESRL is suitable to a large class of models, as the regret bound does not impose a specific form on $f$. Regarding access to $f(\cdot|\mathbf{D}_T)$ for sampling MDPs in real-world problems, as data increases, dependency of results on the prior decreases, so we can use any *working model* to approximate the MDP. Several models are computationally simple to sample from, and can be used for learning. For example, we use the Dirichlet/multinomial, and normal-gamma/normal conjugates for $P^M$ and $R^M$ respectively, which work well for all simulation and real data settings explored in Section 5. In fact, if a Dirichlet prior over the transitions is assumed, the regret bound in Theorem 3.4 can be improved. Chosen priors should be flexible enough to capture the dynamics and easy to sample from efficiently. Next we consider how to discern whether ESRL, or any other fixed policy, is an improvement on the behavior policy.

## 4 Off-Policy Policy Evaluation and Uncertainty Estimation

We now illustrate how the ESRL framework can be used to construct efficient point estimates of the value function, and their posterior distributions. Hypothesis testing can also be used to assess whether the difference in value of two policies is statistically significant (i.e. $\mu^\alpha$ vs. $\pi$).

To compute the estimated value of a given policy $\tilde{\mu}$, we sample $K$ models from the posterior and navigate $M_k$ using $\tilde{\mu}$. This yields samples $V_{\tilde{\mu},1}^{(k)} \sim f_V(\cdot|\mathbf{D}_T)$. We estimate $\mathbb{E}\left[V_{\tilde{\mu},1}^{M^*}(s)|\mathbf{D}_T\right]$ with $\hat{V}_{\tilde{\mu}} = \frac{1}{K}\sum_{k=1}^{K} V_{\tilde{\mu},1}^{(k)}$. Note that we average over the initial states as well, as we are interested to know the marginal value of the policy. A conditional value of the policy function $V_{\tilde{\mu},1}^{M^*}(s_0)$ can also be computed simply by starting all samples at a fixed state $s_0$.

**Theorem 4.1.** *Let* $\tilde{\mu} : \mathcal{S} \times \{1,\ldots,\tau\} \mapsto \mathcal{A}$ *be a pre-specified policy,*

$$\mathbb{E}\left[\hat{V}_{\tilde{\mu}}\middle|\mathbf{D}_T\right] = \mathbb{E}\left[V_{\tilde{\mu},1}^{M^*}(s)\middle|\mathbf{D}_T\right], \hat{V}_{\tilde{\mu}} - \mathbb{E}\left[V_{\tilde{\mu},1}^{M^*}(s)\middle|\mathbf{D}_T\right] = O_p\left(K^{-\frac{1}{2}}\right).$$

Theorem 4.1 ensures that we are indeed estimating the quantity of interest. It establishes that $\hat{V}_{\tilde{\mu}}$ is consistent and unbiased for $\sum_{s\in\mathcal{S}} P_0(s) V_{\tilde{\mu},1}^{M^*}(s)$. As MDP $M^*$ is stochastic, point estimates without measures of uncertainty are not sufficient to evaluate the quality of a policy. For example in an application such as healthcare, there might be policies for which the second best action (treatment) is not significantly different in terms of value, but has less associated secondary risks. Including a secondary risk directly into the method might force us to make strong modeling assumptions. Therefore, testing whether such policies yield a statistically significant difference in value is important. With this information, one can devise a policy that always chooses the safest action (e.g. in clinical terms) and if this yields an equivalent value to the optimal policy, then it is preferable.

**Policy-level hypothesis testing.** Define the value function null hypothesis for two fixed policies $\tilde{\mu}_1, \tilde{\mu}_2$ as the event in which policy $\tilde{\mu}_1$ has a higher expected value than $\tilde{\mu}_2$ conditional on $\mathbf{D}_T$: $H_0 : \mathbb{E}_{s\sim P_0,M^*}\left[V_{\tilde{\mu}_1,1}(s)|\mathbf{D}_T\right] > \mathbb{E}_{s\sim P_0,M^*}\left[V_{\tilde{\mu}_2,1}(s)|\mathbf{D}_T\right]$. The probability of the null under the true model $M^*$ is

$$\mathbb{P}_\mu\left(H_0|\mathbf{D}_T\right) = \sum_{s\in\mathcal{S}} P_0(s)\mathbb{P}\left(V_{\tilde{\mu}_1,1}^{M^*}(s) > V_{\tilde{\mu}_2,1}^{M^*}(s)\middle|s,\mathbf{D}_T\right).$$

We use samples $V_{\tilde{\mu}_\ell}^{(k)}$, $\ell = 1,2$ to estimate the probability of the null with $\hat{\mathbb{P}}_\mu\left(H_0|\mathbf{D}_T\right) = \frac{1}{K}\sum_{k=1}^{K} I\left(V_{\tilde{\mu}_1,1}^{(k)}(s) > V_{\tilde{\mu}_2,1}^{(k)}(s)\right)$. Consistency of this estimator is shown in the Appendix C.2.

## 5 Experiments and Application

We perform several analyses to assess ESRL policy learning, sensitivity to the risk aversion parameter $\alpha$, value function estimation, and finally illustrate how we can interpret the posteriors within the context of the application. The code for implementing ESRL with detailed comments is publicly available[1]. We use the Riverswim environment [28], and a Sepsis data set built from MIMIC-III

data [29]. We compare ESRL to several methods: a) a *naive* baseline made from an ensemble of $K$ DQN models (DQNE), where we simply use the mean for selecting actions, this benchmark is meant to shed light into the empirical benefit of the hypothesis testing in ESRL. b) We argue ESRL can deviate from the behavior policy when allowed by the hypothesis testing, for further investigating the benefit of hypothesis testing, we implement behavior cloning (BC). c) We explore Batch Constrained deep Q-learning (BCQ) which uses regularization towards the behavior policy for offline RL [18, 20, 30]. d) Finally, we also implement a strong benchmark which leverages ensembles and uncertainty estimation in the context of offline RL using random ensembles (REM) [6]. For Riverswim we use 2-128 unit layers, for Sepsis we use 128, 256 unit layers respectively [31]. For ESRL, we use conjugate Dirichlet/multinomial, and normal-gamma/normal for the prior and likelihood of the transition and reward functions respectively.

## 5.1 Riverswim

The Riverswim environment [28] requires deep exploration for achieving high rewards. There are 6 states and two actions: swim right or left. Only swimming left is always successful. There are only two ways to obtain rewards: swimming left while in the far left state will yield a small reward (5/1000) w.p. 1, swimming right in the far right state will yield a reward of 1 w.p. 0.6. The episode lasts 20 time points. We train policy $\pi^0$ using PSRL [16] for 10,000 episodes, we then generate data set $\boldsymbol{D}_T$ with $\pi$, varying both size $T$ and noise $\epsilon$. The offline trained policies are then tested on the environment for 10,000 episodes. This process is repeated 50 times.

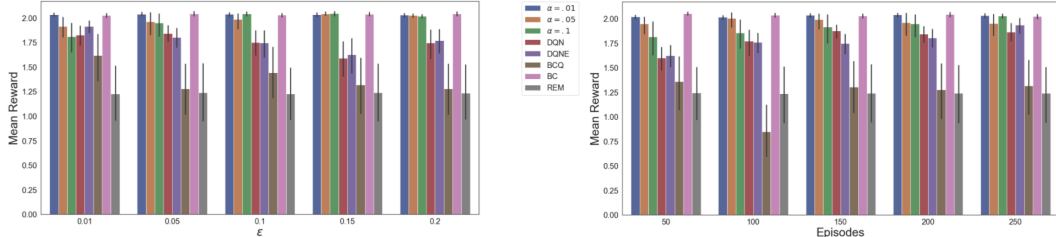

(a) Mean reward for $T$=200 episodes, while varying $\epsilon$-greedy behavior policy $\pi$.

(b) Mean reward for a $\epsilon = 0.05$ in the behavior policy, while varying number of episodes $T$ in $\boldsymbol{D}_T$.

Figure 1: Mean test reward per episode for policies trained offline with ESRL ($\alpha = 0.01, 0.05, 0.1$), DQN, DQNE, BC, BCQ, and REM on Riverswim. Optimal policy expected reward is 2.

**Policy Learning.** We first assess ESRL on Riverswim. The training set sample size $T$ is kept low to make it hard to completely learn the dynamics of the environment. We train an offline policy using ESRL with different risk aversion parameters ($\alpha = 0.01, 0.05, 0.1$). Figure 1 (a) shows mean reward for $T = 200$ episodes while varying $\epsilon$. ESRL proves to be robust to the behavior policy quality. This is expected as when $\epsilon$ is low the environment is not fully explored. This yields high variance in the $Q$ posteriors, which leads ESRL to reject the null more often and favor the behavior policy. For low quality data generating policies there is greater exploration of the environment, which yields narrower posterior distributions for the $Q$ function posteriors, leading ESRL to reject the null when it is indeed beneficial to do so. When behavior policy is almost deterministic, the smaller risk aversion parameter $\alpha$ seems to yield good results as ESRL almost always imitates the behavior policy. BC does well as it seems to estimate the expert behavior well enough regardless of the noise level. Overall $Q$-learning methods lack enough data to learn a good policy. Figure 1 (b) compares methods on an almost constant behavior policy ($\epsilon = 0.05$), so there is little exploration in $\boldsymbol{D}_T$. ESRL is robust as wide posteriors keep it from deviating from $\pi$. Methods other than BC generally fail likely to lack of exploration in $\boldsymbol{D}_T$. However note that in real world data $\pi^0$ is not necessarily optimal, in which case BC will likely not perform very well relative to ESRL or others if there is a high-noise expert policy, which yields a well explored MDP, this is the case in the Sepsis results shown in Figure 4 (c). Finally it's worth noting that REM does better than DQNE in Riverswim but not on Sepsis, we believe this is because the DQN neural networks are smaller, REM outperforms DQNE in a more complex and higher variance setting with more training data such as the Sepsis setting in Section 5.2.

Figure 2 shows Mean Squared Error (MSE) and 95% confidence bands for value estimation of an ESRL policy using $\boldsymbol{D}_T$ while varying $T$. We compare it with sample-based estimates: step importance sampling (IS), and step weighted IS (WIS), and model based estimates which use a full

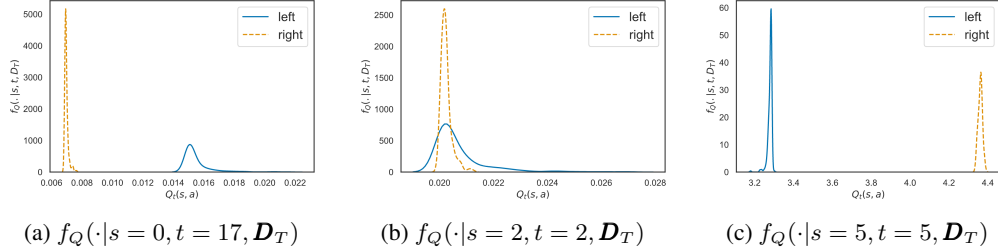

(a) $f_Q(\cdot|s=0,t=17,\boldsymbol{D}_T)$      (b) $f_Q(\cdot|s=2,t=2,\boldsymbol{D}_T)$      (c) $f_Q(\cdot|s=5,t=5,\boldsymbol{D}_T)$

Figure 3: Posterior distributions of $Q_t(s,a)$ functions for fixed $(s,t)$. We use $K=250$ MDP samples. Observed data, $\boldsymbol{D}_T$ has $T=1000$ episodes, generated with $\epsilon=0.2$.

non parametric model (NPM), and an NPM ensemble (NPME). The non parametric models compute the rewards and transition probability tables based on observed counts. The policy is evaluated by using the tables as an MDP model where states are drawn using the estimated transition probability matrix. NPM uses 1000 episodes to evaluate a policy, NPME is an average over 100 NPM estimates. In small data sets ESRL performs substantially better as it uses the model posteriors to overcome rarely visited states in $\boldsymbol{D}_T$. Eventually the priors (which are miss-specified for some state-action pairs) loose importance and ESRL converges to the non-parametric estimates. Sample based estimates are consistently less efficient but converge to the true policy with enough data.

**Hypothesis testing and interpretability with $Q$ function posterior distributions.** We illustrate interpretability of the ESRL method in Riverswim as it is a simple, intuitive setting. Figure 3 shows 3 $Q$ function posterior distributions $f_Q(\cdot|s,t,\boldsymbol{D}_T)$, each for a fixed state-time pair $(s,t)$. Display (a) shows $Q$ functions for the far left state and an advanced time point $t=17$. There is high certainty (no overlap in posteriors) that swimming left will yield a higher reward, as left is successful w.p. 1. $Q_{17}(0,left)$ has a wider posterior as this $(s,a)$ is not common in $\boldsymbol{D}_T$. Display (b) is the most interesting, it sheds light into the utility of uncertainty measures. A naive RL method that only considers mean values, would choose the optimal action according to $\mu$: swimming left. However, there is high

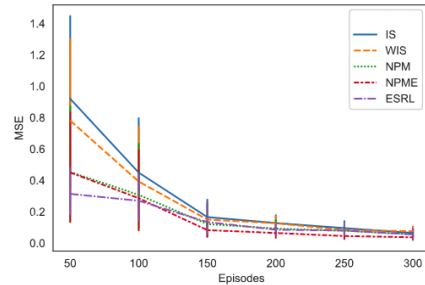

Figure 2: Mean squared error and 95% confidence bands for OPPE of an ESRL policy. We compare step importance sampling (IS), step weighted IS (WIS), a non parametric model (NPM), an NPM ensemble (NPME) and ESRL estimation.

uncertainty associated with such a choice. In fact, we know that the optimal strategy in Riverswim is $\pi(2,2)=right$, hypothesis testing will fail to reject the null and use the behavior action which will lead to a higher expected reward. Display (c) shows $Q$ posteriors for the state furthest to the right, at $t=5$. Choosing right will be successful with high certainty: narrow $Q_5(5,left)$ posterior. Swimming left will still yield a relatively high reward as in the next time point the agent will proceed with the optimal policy (choosing right). As there is no overlap in (a) and (c), the best choice is clear as would be reflected with a hypothesis test.

## 5.2 Sepsis.

We further test ESRL on a Sepsis data set built from MIMIC-III [29]. Sepsis is a state of infection where the immune system gets overwhelmed and can cause tissue damage, organ failure, and death. Deciding treatments and medication dosage is a dynamic and highly challenging task for the clinicians. We consider an action space representing dosage of intravenous fluids for hypovolemia (IV fluids) and vasopressors to counteract vasodilation. The action space $\mathcal{A}$ is size 25: a $5 \times 5$ matrix over discretized dose of vasopressors and IV fluids. The state space is composed of 1,000 clusters estimated using K-means on a 46-feature space which contains measures of the patient's physiological state. We used negative SOFA score as a reward [31], we transform it to be between 0 and 1. The data set used has 12,991 episodes of 10 time steps- measurements every 4-hour interval. We used 80% of episodes for training and 20% for testing.

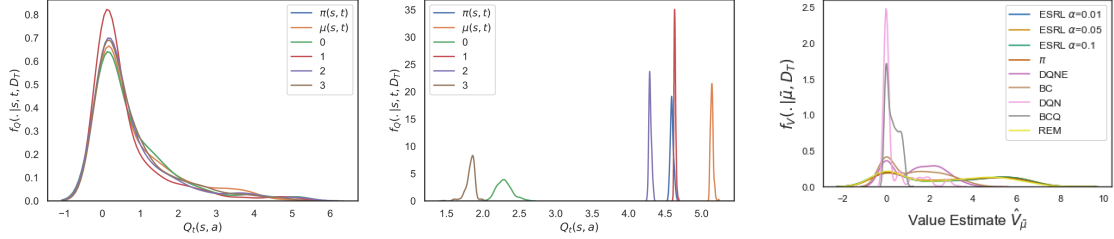

(a) $Q$ function posterior distributions for $(s, t) = (90, 7)$, $a \in \{0, 1, 2, 3, \pi(90, 7), \mu(90, 7)\}$.

(b) $Q$ function posterior distributions for $(s, t) = (5, 8)$, $a \in \{0, 1, 2, 3, \pi(5, 8), \mu(5, 8)\}$.

(c) Posterior distribution for $\hat{V}_{\hat{\mu}}$, for $\pi$, ESRL, BC, BCQ, DQN, DQNE, REM.

Figure 4: Display (a) & (b) show posterior distributions of $Q$ functions at fixed $(s, t)$. Display (c) shows posteriors $\hat{V}$ for policies: $\pi$ and $\mu^\alpha$ for $\alpha = 0.01, 0.05, 0.1$, DQN, DQNE, BC, BCQ and REM for $K = 500$.

Figure 4 (a) & (b) show posterior distributions for two different $(s, t)$ pairs in the Sepsis data set hand-picked to illustrate interpretability. For simplicity we restrict to show the best action: $\mu(s, t)$, physician's action $\pi(s, t)$, and four other low dose actions $a \in \{0, \dots, 3\}$. Display (a) shows posterior distributions over a state rarely observed in $\boldsymbol{D}_T$, hence the $Q$ functions have relatively high standard errors. The expected cumulative inverse SOFA value for this state seems to be relatively stable no matter what action is taken. The $Q$ posteriors for $\mu$ and $\pi$ are practically overlapping so there's no reason to deviate from $\pi$, this is encoded into $\mu^\alpha$ through hypothesis testing. Interpertability is useful in these cases, as a physician might see there is no difference in actions: all will yield similar SOFA scores. Therefore, an action can be chosen to lower risk of side effects. Display (b) on the other hand shows a common state in $\boldsymbol{D}_T$: the low standard errors allow the policy to deviate from $\pi$ at any $\alpha$ level. Within this state, actions $\pi$ and $\mu$ are usually selected so the posteriors for their $Q$ functions are narrow, as opposed to those for $a = 0, 3$. These actions are not prevalent in $\boldsymbol{D}_T$ as they seem to be sub-optimal, so they are less often chosen by doctors and seen in $\boldsymbol{D}_T$.

Figure 4 (c) shows the posterior distribution of the Sepsis value function for different policies. There seems to be a bi-modal distribution: it is easier to control the SOFA scores for patients in the set of states shown in the right mode of the distribution. Physicians know how to do this well as shown by the posterior value function for $\pi$; and ESRL picks up on this. The other clusters of states in the left mode seem to be harder to control. We can appreciate how deviating from the physician's policy is strikingly damaging to the expected value on the test set. DQN and BCQ, DQNE and BC generalize better but under preform relative to ESRL and REM. The $\boldsymbol{D}_T$ is probably not enough to generalize to the test set due to the high dimensional state and action spaces. ESRL through hypothesis testing captures this and hardly deviates from the behavior policy. Thus, it is clear that we cannot do better than $\pi$ given the information in the data, but the posterior suggest the need to learn safe policies as we can do substantially worse with methods that don't account for uncertainty and safety.

## 6 Conclusion

We propose an Expert-Supervised RL (ESRL) approach for offline learning based on Bayesian RL. This framework can learn safe policies from observed data sets. It accounts for uncertainty in the MDP and data logging process to assess when it is safe and beneficial to deviate from the behavior policy. ESRL allows for different levels of risk aversion, which are chosen within the application context. We show a $\tilde{\mathcal{O}}(\tau S \sqrt{AT})$ Bayesian regret bound that is independent of the risk aversion level tailored to the environment and noise level in the data set. The ESRL framework can be used to obtain interpretable posterior distributions for the $Q$ functions and for OPPE. These posteriors are flexible to account for any possible policy function and are amenable to interpretation within the context of the application. An important limitation of ESRL is that it cannot readily handle continuous state spaces which are common in real world applications. Another extension we are interested in is in exploring the comparison of credible intervals as opposed to the null probability estimates. We believe ESRL is a step towards bridging the gap between RL research and real-world applications.

## Broader Impact

We believe ESRL is a tool that can help bring RL closer to real-world applications. In particular this will be useful in the clinical setting to find optimal dynamic treatment regimes for complex diseases, or at least assist in treatment decision making. This is because ESRL's framework lends itself to be questioned by users (physicians) and sheds light into potential biases introduced by the data sampling mechanism used to generate the observed data set. Additionally, using hypothesis testing and accommodating different levels of risk aversion makes the method sensible to offline settings and different real-world applications. It is important when using ESRL and any RL method, to question the validity of the policy's decisions, the quality of the data, and the method that was used to derive these.

## Acknowledgments and Disclosure of Funding

We thank Eric Dunipace for great discussions on Bayesian hypothesis testing and the reviewers for thoughtful feedback, especially regarding state-of-the-art benchmark methods. Funding in support of this work is in part provided by Boehringer Ingelheim Pharmaceuticals. Leo A. Celi is funded by the National Institute of Health through NIBIB R01 EB017205.

## Footnotes

[1]https://github.com/asonabend/ESRL

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
