[Supplementary Material · ESRL. Appendix.pdf]

## Supplementary Material for Expert-Supervised Reinforcement Learning for Offline Policy Learning and Evaluation

## Appendix A  Off-Policy Policy Evaluation and Uncertainty Estimation

In this Section, we follow the lines of Section 4 in the main text with more discussion. We show an Algorithm that collects the ideas presently discussed and an additional Lemma regarding the convergence of the null probability estimator.

We leverage $f(\cdot|\boldsymbol{D}_T)$ to estimate the value function for any policy, and use hypothesis testing for whether there is a meaningful difference in two policy functions (i.e. $\mu^\alpha$ vs. $\pi$). Recall, we compute the estimated value of a given policy $\tilde{\mu}$, by sampling $K$ models from the posterior and navigating $M_k$ using $\tilde{\mu}$ to obtain $V_{\tilde{\mu},1}^{M_k} \sim f_V(\cdot|\boldsymbol{D}_T)$. We estimate $\mathbb{E}\left[V_{\tilde{\mu}}^{M^*}|\boldsymbol{D}_T\right]$ with $\hat{V}_{\tilde{\mu}} = \frac{1}{K}\sum_{k=1}^K V_{\tilde{\mu},1}^{(k)}$. This process is shown in Algorithm 2.

---

**Algorithm 2:** Value function estimation

---

**for** $k = 1,\ldots,K$ **do**

    Set $V_0^{(k)} \leftarrow 0$;

    Sample $M_k \sim f(\cdot|\boldsymbol{D}_T)$, $k = 1,\ldots,K$;

    Sample $s \sim P_0^{M_k}$;

    **for** $t = 1,\ldots,\tau$ **do**

        $a \leftarrow \tilde{\mu}(s,t)$;

        $V_t^{(k)} \leftarrow V_{t-1}^{(k)} + \bar{R}^{M_k}(s,a)$;

        Sample $s' \sim P_a^{M_k}(s'|s)$;

        Set $s \leftarrow s'$;

    **end**

    Set $V_{\tilde{\mu},1}^{(k)} \leftarrow V_\tau^{(k)}$;

**end**

---

Note that we average over the initial states as well, as we are interested to know the marginal value of the policy. A conditional value of the policy function $V_{\tilde{\mu},1}^{M^*}(s)$ can also be computed simply by starting all samples at a fixed state. Analogous to Section 3, we use samples $\left\{V_{\tilde{\mu},1}^{(k)}\right\}_{k=1}^K$ to define a $(1-\alpha)$ CI using the $\alpha$ and $1-\alpha$ quantiles. Note that for policies which are very different from the behavior policy, the posterior distribution will have wider CIs due to the wide distribution shift. This signals that there is not enough information in $\boldsymbol{D}_T$ for the rarely visited state-action pairs $(s,a)$. This happens with OPPE importance sampling estimators as well [2]. As opposed to only considering point estimators of the value function, these CI help to assess whether the estimated value is likely to be accurate or if the estimate is unreliable given the information in $\boldsymbol{D}_T$. Importance sampling based estimators reflect this large distribution shift in high variance estimators.

**Policy-level hypothesis testing.** We use Algorithm 2 to assess whether there is a statistically significant difference in value from two different policies. Define the value function null hypothesis for two fixed policies $\tilde{\mu}_1, \tilde{\mu}_2$ as the event in which policy $\tilde{\mu}_1$ has a higher expected value than $\tilde{\mu}_2$ conditional on $\boldsymbol{D}_T$: $H_0 : \mathbb{E}_{s\sim P_0,M^*}\left[V_{\tilde{\mu}_1}(s)|\boldsymbol{D}_T\right] > \mathbb{E}_{s\sim P_0,M^*}\left[V_{\tilde{\mu}_2}(s)|\boldsymbol{D}_T\right]$. The probability of the null under the true model $M^*$ is

$$\mathbb{P}_\mu^*\left(H_0|\boldsymbol{D}_T\right) = \mathbb{P}\left(V_{\tilde{\mu}_1}^{M^*}(s) > V_{\tilde{\mu}_2}^{M^*}(s)\bigg|\boldsymbol{D}_T\right) = \sum_{s\in\mathcal{S}} P_0(s)\mathbb{P}\left(V_{\tilde{\mu}_1}(s) > V_{\tilde{\mu}_2}(s)\bigg|s,\boldsymbol{D}_T\right).$$

We use the following estimator from samples generated from Algorithm 2:

$$\hat{\mathbb{P}}_\mu\left(H_0|\boldsymbol{D}_T\right) = \frac{1}{K}\sum_{k=1}^K I\left(V_{\tilde{\mu}_1}^{M_k}(s) - V_{\tilde{\mu}_2}^{M_k}(s) > 0\right). \tag{2}$$

**Lemma A.1.** *Let $\mu_1, \mu_2 : \mathcal{S} \times \{1,\ldots,\tau\}$ be two pre-specified policy functions, and let $\hat{P}_\mu(H_0|\boldsymbol{D}_T)$ be defined as in* (2),

$$\hat{\mathbb{P}}_\mu\left(H_0|\boldsymbol{D}_T\right) - \mathbb{P}_\mu\left(H_0|\boldsymbol{D}_T\right) = O_p\left(K^{-\frac{1}{2}}\right),$$

Lemma A.1 ensures consistency of the probability of the null-hypothesis for the value function testing.

## Appendix B   Supporting Lemma

**Lemma B.1.** *(Lemma 1 in [16]) If $f$ is the distribution of $M^*$ then, for any $\sigma(\boldsymbol{D}_T)-$measurable function $g$, and model $M_k \sim f(\cdot|\boldsymbol{D}_T)$:*

$$\mathbb{E}\left[g(M^*)|\boldsymbol{D}_T\right] = \mathbb{E}\left[g(M_k)|\boldsymbol{D}_T\right].$$

## Appendix C   Proof of results in main body

### C.1   Theorem 3.4

In this Subsection we develop the necessary definitions and lemmas, and eventually go on to prove Theorem 3.4. To simplify notation let $\mathbb{P}^*(H_0) \equiv \mathbb{P}^*(H_0|s,t,\boldsymbol{D}_T)$ and $\hat{\mathbb{P}}(H_0) \equiv \hat{\mathbb{P}}(H_0|s,t,\boldsymbol{D}_T)$. Given the behavior policy as defined in Algorithm 1 and the optimal policy under the true MDP $M^*$, we can write the ESRL policy obtained from any $M_k$ sample from Algorithm 1, and it's equivalent version under $M^*$ as:

$$\mu_k^\alpha(s,t) = I\left(\hat{\mathbb{P}}(H_0) < \alpha\right)\mu_k(s,t) + I\left(\hat{\mathbb{P}}(H_0) \geq \alpha\right)\pi(s,t),$$

$$\mu^{\alpha*}(s,t) = I\left(\mathbb{P}^*(H_0) < \alpha\right)\mu^*(s,t) + I\left(\mathbb{P}^*(H_0) \geq \alpha\right)\pi(s,t),$$

we show our result is true for any $\mu_k^\alpha$ and thus it's true for the ESRL policy $\mu^\alpha$. Next we define the policy $\mu_k^\alpha$ which uses the true null probabilities and $\mu_k$ as:

$$\mu_k^{\alpha*}(s,t) = I\left(\mathbb{P}^*(H_0) < \alpha\right)\mu_k(s,t) + I\left(\mathbb{P}^*(H_0) \geq \alpha\right)\pi(s,t).$$

finally let

$$\Delta_i^\mu = \sum_{s \in \mathcal{S}} P_0(s)\left(V_{\mu_k^{\alpha*},1}^{M^*}(s) - V_{\mu_k^\alpha,1}^{M^*}(s)\right)$$

$$\Delta_i^* = \sum_{s \in \mathcal{S}} P_0(s)\left(V_{\mu_k^{\alpha*},1}^{M_k}(s) - V_{\mu_k^{\alpha*},1}^{M^*}(s)\right).$$

Consider function $g : M \mapsto V_{\mu^{\alpha}*,1}^M$, $g$ is $\sigma(\boldsymbol{D}_T)$ measurable for a fixed $\alpha \in [0,1]$ as $\pi(s,t)$, $\mathbb{P}^*(H_0)$ are fixed $\forall (s,t) \in \mathcal{S} \times \{1,\ldots,\tau\}$, thus, by Lemma B.1 for any $M_k \sim f(\cdot|\boldsymbol{D}_T)$

$$\mathbb{E}\left[V_{\mu_k^{\alpha*},1}^{M_k}(s)|\boldsymbol{D}_T\right] = \mathbb{E}\left[V_{\mu^{\alpha*},1}^{M^*}(s)|\boldsymbol{D}_T\right],$$

now using iterated expectations we get $\mathbb{E}\left[V_{\mu_k^{\alpha*},1}^{M_k}(s)\right] = \mathbb{E}\left[V_{\mu^{\alpha*},1}^{M^*}(s)\right]$.

We use this to re-express the expected regret for episode $i$ under model $k$ computed with Algorithm 1 as

$$\mathbb{E}[\Delta_i] = \mathbb{E}\left[\sum_{s \in \mathcal{S}} P_0(s)\left(V_{\mu^{\alpha*},1}^{M^*}(s) - V_{\mu_k^\alpha,1}^{M^*}(s)\right)\right]$$

$$= \sum_{s \in \mathcal{S}} P_0(s)\left(\mathbb{E}\left[V_{\mu^{\alpha*},1}^{M^*}(s)\right] - \mathbb{E}\left[V_{\mu_k^\alpha,1}^{M^*}(s)\right]\right)$$

$$= \sum_{s \in \mathcal{S}} P_0(s)\left(\mathbb{E}\left[V_{\mu_k^{\alpha*},1}^{M_k}(s)\right] - \mathbb{E}\left[V_{\mu_k^\alpha,1}^{M^*}(s)\right]\right)$$

$$= \mathbb{E}[\Delta_i^*] + \mathbb{E}[\Delta_i^\mu],$$

where the last step follows from adding and subtracting $\mathbb{E}\left[V_{\mu_k^{\alpha*},1}^{M^*}(s)\right]$.

We first consider $\mathbb{E}\left[\Delta_i^*\right]$, we use a strategy similar to [16], but do not make an *iid* assumption for within-episode observations. Define the following Bellman operator $\mathcal{T}_{\mu^\alpha}^M$ for any MDP $M$, policy $\mu^\alpha$, and value function $V$ to be

$$\mathcal{T}_{\mu^\alpha}^M V(s) = \bar{R}^M(s, \mu^\alpha(s,t)) + \sum_{s' \in \mathcal{S}} P_{\mu^\alpha(s,t)}^M(s'|s)V(s'), \tag{3}$$

this lets us write $V_{\mu^\alpha,t}^M(s) = \mathcal{T}_{\mu^\alpha}^M V_{\mu^\alpha,t+1}^M(s)$.

The next Lemma will let us express term $\mathbb{E}\left[\Delta_i^* \middle| M^*, M_k\right]$ in terms of the Bellman operator.

**Lemma C.1.** *If $f$ is the distribution of $M^*$, then*

$$\mathbb{E}\left[\Delta_i^* \middle| M^*, M_k\right] = \mathbb{E}\left[\sum_{j=1}^{\tau}\left(\mathcal{T}_{\mu_k^{\alpha*}(\cdot,j)}^{M_k} - \mathcal{T}_{\mu_k^{\alpha*}(\cdot,j)}^{M^*}\right)V_{\mu_k^{\alpha*},j+1}^{M_k}(s_{j+1})\middle| M^*, M_k\right].$$

We now define a confidence set for the reward and transition estimated probabilities.

**Lemma C.2.** *Let $\mathcal{I}$ denote the set of index $i, j$ for episodes in $\boldsymbol{D}_T = \{(s_{i1}, a_{i1}, r_{i1}, \ldots, s_{i\tau}, a_{i\tau}, r_{i\tau})\}_{i=1}^T$, that is: $\mathcal{I} = \left\{(i,j)\middle| i \in \{1,\ldots,T\}, j \in \{1,\ldots,\tau\}\right\}$. Further let $N_T(s,a)$ be the number of times $(s,a)$ was sampled in $\boldsymbol{D}_T$: $N_T(s,a) = \sum_{i,j \in \mathcal{I}} I(S_{ij} = s, A_{ij} = a)$, let $\hat{P}_a(\cdot|s)$ and $\hat{R}(s,a)$ be non-parametric estimators for the distribution of transitions and rewards observed after sampling $T$ episodes:*

$$\hat{P}_a(s'|s) = \frac{\sum_{i,j \in \mathcal{I}} I(s_{i,j+1} = s')I(s_{ij} = s, a_{ij} = a)}{N_T(s,a)}, \quad \hat{R}(s,a) = \frac{\sum_{ij \in \mathcal{I}} I(s_{ij} = s, a_{ij} = a)r_{ij}}{N_T(s,a)}.$$

*Define the confidence set:*

$$\mathcal{M}_T \equiv \left\{M : \left\|\hat{P}_a(\cdot|s) - P_a^M(\cdot|s)\right\|_1 \leq \beta_T(s,a), \left|\hat{R}(s,a) - R^M(s,a)\right|_1 \leq \beta_T(s,a) \; \forall(s,a)\right\},$$

*where $\beta_T(s,a) \equiv \frac{\sqrt{8ST\log(2SAT)}}{\max\{1, N_T(s,a)\}}$, then $P\left(M^* \notin \mathcal{M}_T\right) < \frac{1}{T}$.*

*Proof of Theorem 3.4.* We start by summing $\Delta_i^*$ over all episodes:

$$\mathbb{E}\left[\sum_{i=1}^T \Delta_i^*\right] \leq \mathbb{E}\left[\sum_{i=1}^T \Delta_i^* I(M_k, M^* \in \mathcal{M}_T)\right] + \tau \sum_{i=1}^T \left(\mathbb{P}(M_k \notin \mathcal{M}_T) + \mathbb{P}(M^* \notin \mathcal{M}_T)\right)$$

$$\leq \mathbb{E}\left[\mathbb{E}\left[\sum_{i=1}^T \Delta_i^* | M_k, M^*\right] I(M_k, M^* \in \mathcal{M}_k)\right] + 2\tau$$

$$\leq \mathbb{E}\left[\sum_{i=1}^T \sum_{j=1}^{\tau}\left|\left(\mathcal{T}_{\mu_k^{\alpha*}(\cdot,j)}^{M_k} - \mathcal{T}_{\mu_k^{\alpha*}(\cdot,j)}^{M^*}\right)V_{\mu_k^{\alpha*},j+1}^{M_k}(s_j)\right| I(M_k, M^* \in \mathcal{M}_k)\right] + 2\tau$$

where the first step follows by conditioning on event $I(M_k \in \mathcal{M}_T, M^* \in \mathcal{M}_T)$ and it's complement, and from the fact that $\Delta_i^* \leq \tau$ as all rewards $R(s,a) \in [0,1]$. The second step follows from iterated expectations and Lemma C.2 as $\mathbb{P}[I(M^* \notin \mathcal{M}_T)] \leq \frac{1}{T}$. Also since $\mathcal{M}_T$ is a $\sigma(D_T)$-measurable function by Lemma B.1 we have $\mathbb{E}\left[I\left(M_k \notin \mathcal{M}_T\right)|D_T\right] = \mathbb{E}\left[I\left(M^* \notin \mathcal{M}_T\right)|D_T\right]$, using iterated expectations we have $\mathbb{P}[I(M_k \notin \mathcal{M}_T)] \leq \frac{1}{T}$. The last step follows from Lemma C.1. Next using (3)

the last equation can be re-written as

$$\mathbb{E}\left[\sum_{i=1}^{T}\sum_{j=1}^{\tau} I\left(\mathbb{P}^*(H_0) \geq \alpha\right)\left\{\bar{R}^{M_k}(s, \pi(s,j)) - \bar{R}^{M^*}(s, \pi(s,j))\right\} I(M_k, M^* \in \mathcal{M}_k)\right]$$

$$+ \mathbb{E}\left[\sum_{i=1}^{T}\sum_{j=1}^{\tau} I\left(\mathbb{P}^*(H_0) \geq \alpha\right)\left\{\sum_{s' \in \mathcal{S}}\left|P_{\pi(s,j)}^{M_k}(s'|s) - P_{\pi(s,j)}^{M^*}(s'|s)\right| V_{\mu_k^{\alpha*},j+1}^{M_k}(s_{j+1})\right\} I(M_k, M^* \in \mathcal{M}_k)\right]$$

$$+ \mathbb{E}\left[\sum_{i=1}^{T}\sum_{j=1}^{\tau} I\left(\mathbb{P}^*(H_0) < \alpha\right)\left\{\bar{R}^{M_k}(s, \mu_k(s,j)) - \bar{R}^{M^*}(s, \mu_k(s,j))\right\} I(M_k, M^* \in \mathcal{M}_k)\right]$$

$$+ \mathbb{E}\left[\sum_{i=1}^{T}\sum_{j=1}^{\tau} I\left(\mathbb{P}^*(H_0) < \alpha\right)\left\{\sum_{s' \in \mathcal{S}}\left|P_{\mu_k(s,j)}^{M_k}(s'|s) - P_{\mu_k(s,j)}^{M^*}(s'|s)\right| V_{\mu_k^{\alpha*},j+1}^{M_k}(s_{j+1})\right\} I(M_k, M^* \in \mathcal{M}_k)\right]$$

$$+ 2\tau$$

$$\leq \mathbb{E}\left[\tau\sum_{i=1}^{T}\sum_{j=1}^{\tau} \min\left\{\beta_T(s_{ij}, \pi(s_{ij},j)), 1\right\}\right] + \mathbb{E}\left[\tau\sum_{i=1}^{T}\sum_{j=1}^{\tau} \min\left\{\beta_T(s_{ij}, \mu_k(s_{ij},j)), 1\right\}\right] + 2\tau,$$

where the last step follows by Lemma C.2, next:

$$\leq \mathbb{E}\left[\tau\sum_{i=1}^{T}\sum_{j=1}^{\tau} \frac{\sqrt{8ST\log(2SAT)}}{\min\{N_T(s_{ij}, \mu_k(s_{ij},j))\}}\right] + \mathbb{E}\left[\tau\sum_{i=1}^{T}\sum_{j=1}^{\tau} \frac{\sqrt{8ST\log(2SAT)}}{\min\{1, N_T(s_{ij}, \pi(s_{ij},j))\}}\right] + 2\tau$$

$$\leq M_1\sqrt{\tau^2 SAT} + M_2\tau\sqrt{S^2 AT \log(SAT)} + 2\tau < M_3\tau S\sqrt{AT\log(SAT)} + 2\tau,$$

where the last step follows by Appendix B in [16] with constants $M_1, M_2, M_3$.

We next analyze

$$\mathbb{E}\left[\Delta_i^{\mu}\right] = \sum_{s \in \mathcal{S}} P_0(s)\left(\mathbb{E}\left[V_{\mu_k^{\alpha*},1}^{M^*}(s)\right] - \mathbb{E}\left[V_{\mu_k^{\alpha},1}^{M^*}(s)\right]\right).$$

We can write the second term as

$$\mathbb{E}\left[V_{\mu_k^{\alpha},1}^{M^*}(s)\right] = \mathbb{E}\left[\sum_{j=1}^{\tau} I\left(\hat{\mathbb{P}}(H_0) < \alpha\right) R^{M^*}(s_j, \mu_k(s_j,j)) + I\left(\hat{\mathbb{P}}(H_0) \geq \alpha\right) R^{M^*}(s_j, \pi(s_j,j))\right],$$

we extend the null probability notation to be explicit on the time index: $\mathbb{P}_j^*(H_0) = \mathbb{P}^*(H_0|s_j, j, \boldsymbol{D}_T), \hat{\mathbb{P}}_j(H_0) = \hat{\mathbb{P}}(H_0|s_j, j, \boldsymbol{D}_T)$. By Lemma 3.2, $\exists \delta > 0$ such that $\hat{\mathbb{P}}_j(H_0) - \mathbb{P}_j^*(H_0) \leq \delta \ \forall s \in \mathcal{S}, j \in \{1, \ldots, \tau\}$ with high probability, therefore

$$\begin{aligned}
\mathbb{P}_j^*(H_0) < \alpha - \delta &\implies \mathbb{P}\left(\hat{\mathbb{P}}_j(H_0) < \alpha\right) = 1 - O_p\left(K^{-\frac{1}{2}}\right), \\
\mathbb{P}_j^*(H_0) \geq \alpha + \delta &\implies \mathbb{P}\left(\hat{\mathbb{P}}_j(H_0) \geq \alpha\right) = 1 - O_p\left(K^{-\frac{1}{2}}\right).
\end{aligned} \tag{4}$$

As $\mathcal{I}_1, \mathcal{I}_2$ in Algorithm 1 are mutually exclusive, $\hat{\mathbb{P}}_j(H_0)$ are independent to $\mu_k(s,j) \ \forall s \in \mathcal{S}, j \in \{1, \ldots, \tau\}$, therefore starting with $V_{\mu_k^{\alpha},\tau}^{M^*}(s)$ we have

$$\mathbb{E}\left[V_{\mu_k^\alpha,\tau}^{M^*}(s)\right]$$

$$=I\left(\mathbb{P}_\tau^*(H_0)<\alpha-\delta\right)\left\{\mathbb{E}\left[I\left(\hat{\mathbb{P}}_\tau(H_0)<\alpha\right)\right]\bar{R}^{M^*}(s_\tau,\mu_k(s_\tau,\tau))+\mathbb{E}\left[I\left(\hat{\mathbb{P}}_\tau(H_0)\geq\alpha\right)\right]\bar{R}^{M^*}(s_\tau,\pi(s_\tau,\tau))\right\}$$

$$+I\left(\mathbb{P}_\tau^*(H_0)\geq\alpha-\delta\right)\left\{\mathbb{E}\left[I\left(\hat{\mathbb{P}}_\tau(H_0)<\alpha\right)\right]\bar{R}^{M^*}(s_\tau,\mu_k(s_\tau,\tau))+\mathbb{E}\left[I\left(\hat{\mathbb{P}}_\tau(H_0)\geq\alpha\right)\right]\bar{R}^{M^*}(s_\tau,\pi(s_\tau,\tau))\right\}$$

$$+I\left(\mathbb{P}_\tau^*(H_0)\in[\alpha-\delta,\alpha+\delta)\right)\left\{\mathbb{E}\left[I\left(\hat{\mathbb{P}}_\tau(H_0)<\alpha\right)\right]\bar{R}^{M^*}(s_\tau,\mu_k(s_\tau,\tau))+\mathbb{E}\left[I\left(\hat{\mathbb{P}}_\tau(H_0)\geq\alpha\right)\right]\bar{R}^{M^*}(s_\tau,\pi(s_\tau,\tau))\right\}\bigg]$$

$$=I\left(\mathbb{P}_\tau^*(H_0)<\alpha-\delta\right)\bar{R}^{M^*}(s_\tau,\mu_k(s_\tau,\tau))+O_p\left(K^{-\frac{1}{2}}\right)$$

$$+I\left(\mathbb{P}_\tau^*(H_0)\geq\alpha-\delta\right)\bar{R}^{M^*}(s_\tau,\mu_k(s_\tau,\tau))+O_p\left(K^{-\frac{1}{2}}\right)$$

$$+I\left(\mathbb{P}_\tau^*(H_0)\in[\alpha-\delta,\alpha+\delta)\right)O_p\left(K^{-\frac{1}{2}}\right)$$

$$=\mathbb{E}\left[V_{\mu_k^{\alpha*},\tau}^{M^*}(s)\right]+O_p\left(K^{-\frac{1}{2}}\right),$$

where the first step follows from $\mathcal{I}_1,\mathcal{I}_2$ being independent, the second step follows from (4) and last step from definition of $V_{\mu_k^{\alpha*},\tau}^{M^*}(s)$. Iterating backards from $\tau-1\ldots,1$ and applying the same steps as above we get

$$\mathbb{E}\left[V_{\mu_k^\alpha,1}^{M^*}(s)\right]=\mathbb{E}\left[V_{\mu_k^{\alpha*},1}^{M^*}(s)\right]+O_p\left(\tau K^{-\frac{1}{2}}\right).$$

therefore we have $\mathbb{E}\left[\sum_{i=1}^T\Delta_i^\mu\right]=O_p\left(T\tau K^{-\frac{1}{2}}\right)$, choosing $K=\mathcal{O}(T)$ we get $\mathbb{E}\left[\sum_{i=1}^T\Delta_i^\mu\right]=O_p\left(\sqrt{T}\tau\right)$ which is dominated by $\mathbb{E}\left[\sum_{i=1}^T\Delta_i^*\right]$.

Putting both terms together we have

$$\mathbb{E}\left[\sum_{i=1}^T\Delta_i\right]=\mathbb{E}\left[\sum_{i=1}^T\Delta_i^*\right]+\mathbb{E}\left[\sum_{i=1}^T\Delta_i^\mu\right]=\mathcal{O}\left(\tau S\sqrt{AT\log(SAT)}\right).$$

$\square$

### C.2 Proofs for other results in main body

*Proof of Lemma 3.1.* To establish $\hat{Q}_{\tilde{\mu},t}(s,a)$ is unbiased, note that for any fixed $(t,s,a)$, $M_k\sim f(\cdot|\boldsymbol{D}_T)$ are *iid*, now for a given policy function $\tilde{\mu}$ :

$$\mathbb{E}\left[\hat{Q}_{\tilde{\mu},t}(s,a)\bigg|s,a,t,\boldsymbol{D}_T\right]=\mathbb{E}\left[\frac{1}{K}\sum_{k=1}^K Q_{\tilde{\mu},t}^{(k)}(s,a)\bigg|s,a,t,\boldsymbol{D}_T\right]$$

$$=\frac{1}{K}\sum_{k=1}^K\mathbb{E}\left[Q_{\tilde{\mu},t}^{(k)}(s,a)\bigg|s,a,t,\boldsymbol{D}_T\right]=\mathbb{E}\left[Q_{\tilde{\mu},t}^{M^*}(s,a)\bigg|s,a,t,\boldsymbol{D}_T\right]$$

where the last step follows from Lemma B.1 with $g:M\mapsto Q_{\tilde{\mu},t}^M(s,a)$ which is $\sigma(\boldsymbol{D}_T)-$ measurable.

To establish the rate, we have that $R^M(s,a)\in[0,1]\ \forall(s,a)\in\mathcal{S}\times\mathcal{A}, t=1,\ldots,\tau$ thus $Q_t^{(k)}(s,a)\leq\tau$. By definition $\hat{Q}_t(s,a)-\mathbb{E}\left[Q_{\mu,t}^{M^*}(s,a)\bigg|s,a,t,\boldsymbol{D}_T\right]=O_p\left(K^{-\frac{1}{2}}\right)$ if and only if for any $\epsilon>0$, $\exists M_\epsilon>0$ such that

$$\mathbb{P}\left(\hat{Q}_{\tilde{\mu},t}(s,a)-\mathbb{E}\left[Q_{\tilde{\mu},t}^{M^*}(s,a)\bigg|s,a,t,\boldsymbol{D}_T\right]>K^{-\frac{1}{2}}M_\epsilon\bigg|s,a,t,\boldsymbol{D}_T\right)\leq\epsilon\ \ \forall K.$$

Note that for any $M > 0$,

$$\mathbb{P}\left(\hat{Q}_{\tilde{\mu},t}(s,a) - \mathbb{E}\left[Q_{\tilde{\mu},t}^{M^*}(s,a)\Big|t,s,a,\boldsymbol{D}_T\right] > K^{-\frac{1}{2}}M\Big|t,s,a,\boldsymbol{D}_T\right)$$

$$=\mathbb{P}\left(\frac{1}{K}\sum_{k=1}^{K}Q_{\tilde{\mu},t}^{(k)}(s,a) - \mathbb{E}\left[Q_{\tilde{\mu},t}^{M^*}(s,a)\Big|s,a,t,\boldsymbol{D}_T\right] > K^{-\frac{1}{2}}M\Big|s,a,t,\boldsymbol{D}_T\right)$$

$$\leq \exp\left\{-\frac{2M^2 K^{-1}K^2}{K\tau^2}\right\} = \exp\left\{-\frac{2M^2}{\tau^2}\right\},$$

which follows from Hoeffding's inequality as conditional on $s,a,t,\tilde{\mu}$ and $\boldsymbol{D}_T$, $\left\{Q_{\tilde{\mu},t}^{(k)}(s,a)\right\}_{k=1}^{K}$ are $iid$ with mean $\mathbb{E}\left[Q_{\tilde{\mu},t}^{M^*}(s,a)\Big|s,a,t,\boldsymbol{D}_T\right]$. The result follows from choosing $M_\epsilon > 0$ large enough such that $\exp\left\{-\frac{2M_\epsilon^2}{\tau^2}\right\} < \epsilon$.

$\square$

*Proof of Lemma 3.2.* To simplify notation, let $Z^{(k)} \equiv I\left(Q_{\mu_k^\alpha,t}^{(k)}(s,\mu_k(s,t)) - Q_{\mu_k^\alpha,t}^{(k)}(s,\pi(s,t)) \leq 0\right)$, then by definition $Z^{(k)} - \mathbb{E}\left[Z^{(k)}\right] = O_p\left(K^{-\frac{1}{2}}\right)$ if and only if for any $\epsilon > 0$, $\exists M_\epsilon > 0$ such that

$$\mathbb{P}\left(Z^{(k)} - \mathbb{E}\left[Z^{(k)}\right] > K^{-\frac{1}{2}}M_\epsilon\Big|t,s,\boldsymbol{D}_T\right) \leq \epsilon \quad \forall K.$$

Note that for any $M > 0$,

$$\mathbb{P}\left(\hat{\mathbb{P}}(H_0|t,s,\boldsymbol{D}_T) - \mathbb{E}\left[Z^{(k)}|t,s,\boldsymbol{D}_T\right] > K^{-\frac{1}{2}}M|t,s,\boldsymbol{D}_T\right)$$

$$=\mathbb{P}\left(\frac{1}{K}\sum_{k=1}^{K}Z^{(k)} - \mathbb{E}\left[Z^{(k)}|t,s,\boldsymbol{D}_T\right] > MK^{-\frac{1}{2}}\Big|t,s,\boldsymbol{D}_T\right)$$

$$\leq \exp\left\{-\frac{2M^2 K^{-1}K^2}{K\tau^2}\right\} = \exp\left\{-\frac{2M^2}{\tau^2}\right\},$$

where the inequality follows from Hoeffding's inequality as $\left\{Z^{(k)}\right\}_{k=1}^{K}$ are $iid$ with mean $\mathbb{E}\left[Z^{(k)}\Big|t,s,\boldsymbol{D}_T\right]$, since $\mathcal{I}_1, \mathcal{I}_2$ in Algorithm 1 are disjoint. We can choose $M_\epsilon > 0$ large enough such that $\exp\left\{-\frac{2M^2}{\tau^2}\right\} < \epsilon$. Next note that as $\pi$ is fixed, by Lemma B.1, with $g : M \mapsto I\left(Q_{\mu^\alpha,t}^{M}(s,\mu(s,t)) - Q_{\mu^\alpha,t}^{M}(s,\pi(s,t)) \leq 0\right)$ for any $M_k \sim f(\cdot|\boldsymbol{D}_T)$

$$\mathbb{E}\left[I\left(Q_{\mu_k^\alpha,t}^{(k)}(s,\mu_k(s,t)) - Q_{\mu_k^\alpha,t}^{(k)}(s,\pi(s,t)) \leq 0\right)\Big|t,s,\boldsymbol{D}_T\right]$$

$$=\mathbb{E}\left[I\left(Q_{\mu^{\alpha*},t}^{M^*}(s,\mu^*(s,t)) - Q_{\mu^{\alpha*},t}^{M^*}(s,\pi(s,t)) \leq 0\right)\Big|t,s,\boldsymbol{D}_T\right]$$

$$=\mathbb{P}(H_0|t,s,\boldsymbol{D}_T)$$

which follows from using disjoint sets $\mathcal{I}_1, \mathcal{I}_2$ in Algorithm 1. Substituting this in the probability statement gives us

$$\hat{\mathbb{P}}\left(H_0|t,s,\boldsymbol{D}_T\right) - \mathbb{P}\left(H_0|t,s,\boldsymbol{D}_T\right) = O_p\left(K^{-\frac{1}{2}}\right),$$

which is our required result.

$\square$

*Proof of Theorem 4.1.* We start by showing $\hat{V}_{\tilde{\mu}}$ is unbiased:

$$\mathbb{E}\left[\hat{V}_{\tilde{\mu}}(s)|\boldsymbol{D}_T,\tilde{\mu}\right] = \frac{1}{K}\sum_{k=1}^{K}\mathbb{E}\left[V_{\tilde{\mu},1}^{(k)}(s)\Big|\boldsymbol{D}_T\right].$$

where the first step follows from definition, and the $M_k \sim f(\cdot|\boldsymbol{D}_T)$ being *iid*, now by Lemma B.1 with $g : M \mapsto V_{\tilde{\mu},1}^M$ we have

$$\mathbb{E}\left[\hat{V}_{\tilde{\mu}}|\boldsymbol{D}_T\right] = \mathbb{E}\left[V_{\tilde{\mu},1}^{M^*}(s)|\boldsymbol{D}_T\right].$$

To establish the rate, we have that $V_{\tilde{\mu},1}^{(k)} \leq \tau$ as all rewards are between $[0, 1]$ by definition $\hat{V}_{\tilde{\mu}} - \mathbb{E}\left[V_{\tilde{\mu},1}^{M^*}(s)|\boldsymbol{D}_T\right] = O_p\left(K^{-\frac{1}{2}}\right)$ if and only if for any $\epsilon > 0$, $\exists M_\epsilon > 0$ such that

$$\mathbb{P}\left(\hat{V}_{\tilde{\mu}} - \mathbb{E}\left[V_{\tilde{\mu},1}^{M^*}(s)|\boldsymbol{D}_T\right] > K^{-\frac{1}{2}}M_\epsilon\right) \leq \epsilon \quad \forall K.$$

Note that for any $M > 0$,

$$\mathbb{P}\left(\hat{V}_{\tilde{\mu}} - \mathbb{E}\left[V_{\tilde{\mu},1}^{M^*}(s)|\boldsymbol{D}_T\right] > K^{-\frac{1}{2}}M\right) = \mathbb{P}\left(\frac{1}{K}\sum_{k=1}^K V_{\tilde{\mu},1}^{(k)} - \mathbb{E}\left[V_{\tilde{\mu},1}^{M^*}(s)|\boldsymbol{D}_T\right] > K^{-\frac{1}{2}}M\right)$$

$$\leq \exp\left\{-\frac{2M^2 K^{-1}K^2}{K\tau^2}\right\} = \exp\left\{-\frac{2M^2}{\tau^2}\right\},$$

where the inequality follows from Hoeffding's inequality as $\left\{V_{\tilde{\mu},1}^{(k)}\right\}_{k=1}^K$ are *iid* with mean $\mathbb{E}\left[V_{\tilde{\mu},1}^{M^*}(s)|\boldsymbol{D}_T\right]$. The result follows from choosing $M_\epsilon > 0$ large enough such that $\exp\left\{-\frac{2M^2}{\tau^2}\right\} < \epsilon$. $\qquad\square$

## Appendix D   Proofs for Supplementary results

*Proof of Lemma A.1.* First note that conditional on $\boldsymbol{D}_T$ with $g : M \mapsto I\left(V_{\mu_1}^M(s) - V_{\mu_2}^M(s) > 0\right)$, by Lemma B.1

$$\mathbb{E}\left[I\left(V_{\mu_1}^{M_k}(s) - V_{\mu_2}^{M_k}(s) > 0\right)\bigg|\boldsymbol{D}_T\right] = \mathbb{E}\left[I\left(V_{\mu_1}^{M^*}(s) - V_{\mu_2}^{M^*}(s) > 0\right)\bigg|\boldsymbol{D}_T\right] = \mathbb{P}_\mu\left(H_0|\boldsymbol{D}_T\right)$$

By definition $\hat{\mathbb{P}}_\mu(H_0|\boldsymbol{D}_T) - \mathbb{P}_\mu\left(H_0|\boldsymbol{D}_T\right) = O_p\left(K^{-\frac{1}{2}}\right)$ if and only if for any $\epsilon > 0$, $\exists M_\epsilon > 0$ such that

$$\mathbb{P}\left(\hat{\mathbb{P}}_\mu(H_0|\boldsymbol{D}_T) - \mathbb{P}_\mu\left(H_0|\boldsymbol{D}_T\right) > K^{-\frac{1}{2}}M_\epsilon\bigg|\boldsymbol{D}_T\right) \leq \epsilon \quad \forall K.$$

Now, for any $M > 0$,

$$\mathbb{P}\left(\hat{\mathbb{P}}_\mu(H_0|\boldsymbol{D}_T) - \mathbb{P}_\mu\left(H_0|\boldsymbol{D}_T\right) > K^{-\frac{1}{2}}M_\epsilon\bigg|\boldsymbol{D}_T\right)$$

$$= \mathbb{P}\left(\frac{1}{K}\sum_{k=1}^K I\left(V_{\mu_1,1}^{(k)} - V_{\mu_2,1}^{(k)} > 0\right) - \mathbb{P}_\mu\left(H_0|\boldsymbol{D}_T\right) > MK^{-\frac{1}{2}}\bigg|\boldsymbol{D}_T\right)$$

$$\leq \exp\left\{-\frac{2M^2 K^{-1}K^2}{K\tau^2}\right\} = \exp\left\{-\frac{2M^2}{\tau^2}\right\},$$

where the inequality follows from Hoeffding's inequality as the indicators $\left\{I\left(V_{\mu_1,1}^{(k)} - V_{\mu_2,1}^{(k)} > 0\right)\right\}_{k=1}^K$ are *iid* with mean $\mathbb{P}_\mu\left(H_0|\boldsymbol{D}_T\right)$. We can choose $M_\epsilon > 0$ large enough such that $\exp\left\{-\frac{2M^2}{\tau^2}\right\} < \epsilon$. $\qquad\square$

*Proof of Lemma C.1.* We first write the estimated regret as a sum of difference in value functions and a Bellman error.

I) We'll denote the sequence of states for an episode as $s_1, s_2, \ldots, s_\tau$, define

$$\mathcal{W}_j = \left( \mathcal{T}^{M_k}_{\mu_k^{\alpha*}(\cdot,j)} - \mathcal{T}^{M^*}_{\mu_k^{\alpha*}(\cdot,j)} \right) V^{M_k}_{\mu_k^{\alpha*},j+1}(s_{j+1})$$

$$\mathbb{T}_j = \mathcal{T}^{M^*}_{\mu_k^{\alpha*}(\cdot,j)} \left( V^{M_k}_{\mu_k^{\alpha*},j+1} - V^{M^*}_{\mu_k^{\alpha*},j+1} \right)(s_{j+1})$$

using (3) we can write

$$\left( V^{M_k}_{\mu_k^{\alpha*},1} - V^{M^*}_{\mu_k^{\alpha*},1} \right)(s_1) = \left( \mathcal{T}^{M_k}_{\mu_k^{\alpha*}(\cdot,1)} V^{M_k}_{\mu_k^{\alpha*},2} - \mathcal{T}^{M^*}_{\mu_k^{\alpha*}(\cdot,1)} V^{M^*}_{\mu_k^{\alpha*},2} \right)(s_2)$$

$$= \left( \mathcal{T}^{M_k}_{\mu_k^{\alpha*}(\cdot,1)} V^{M_k}_{\mu_k^{\alpha*},2} - \mathcal{T}^{M^*}_{\mu_k^{\alpha*}(\cdot,1)} V^{M_k}_{\mu_k^{\alpha*},2} + \mathcal{T}^{M^*}_{\mu_k^{\alpha*}(\cdot,1)} V^{M_k}_{\mu_k^{\alpha*},2} - \mathcal{T}^{M^*}_{\mu_k^{\alpha*}(\cdot,1)} V^{M^*}_{\mu_k^{\alpha*},2} \right)(s_2)$$

$$= \mathcal{W}_1 + \mathbb{T}_1,$$

with the same steps we can generalize this to

$$\left( V^{M_k}_{\mu_k^{\alpha*},j} - V^{M^*}_{\mu_k^{\alpha*},j} \right)(s_j) = \mathcal{W}_j + \mathbb{T}_j. \tag{5}$$

Next let

$$e_j = \left( I\left(\mathbb{P}^*(H_0) < \alpha\right) \sum_{s' \in \mathcal{S}} P^{M^*}_{\mu_k(s,j)}(s'|s) + I\left(\mathbb{P}^*(H_0) \geq \alpha\right) \sum_{s' \in \mathcal{S}} P^{M^*}_{\pi(s,j)}(s'|s) \right)$$

$$\times \left( V^{M_k}_{\mu_k^{\alpha*},j+1} - V^{M^*}_{\mu_k^{\alpha*},j+1} \right)(s') - \left( V^{M_k}_{\mu_k^{\alpha*},j+1} - V^{M^*}_{\mu_k^{\alpha*},j+1} \right)(s_{j+1}),$$

using the Bellman operator we get

$$\mathbb{T}_j = \left( V^{M_k}_{\mu_k^{\alpha*},j+1} - V^{M^*}_{\mu_k^{\alpha*},j+1} \right)(s_{j+1}) + e_j,$$

then we can write $\mathbb{T}_1 = \left( V^{M_k}_{\mu_k^{\alpha*},2} - V^{M^*}_{\mu_k^{\alpha*},2} \right)(s_2) + e_1$, with the above definitions and repeated use of (5):

$$\left( V^{M_k}_{\mu_k^{\alpha*},1} - V^{M^*}_{\mu_k^{\alpha*},1} \right)(s_1) = \mathcal{W}_1 + \mathbb{T}_1$$

$$= \mathcal{W}_1 + \left( V^{M_k}_{\mu_k^{\alpha*},2} - V^{M^*}_{\mu_k^{\alpha*},2} \right)(s_2) + e_1$$

$$= \mathcal{W}_1 + \mathcal{W}_2 + \left( V^{M_k}_{\mu_k^{\alpha*},3} - V^{M^*}_{\mu_k^{\alpha*},3} \right)(s_3) + e_1 + e_2$$

$$\vdots$$

$$= \sum_{j=1}^{\tau} \mathcal{W}_j + e_j.$$

II) Next we consider $\mathbb{E}\left[e_j | M_k, M^*\right]$:

$$\mathbb{E}\left[ e_j \Big| M_k, M^* \right]$$

$$= \mathbb{E}\left[ I\left(\mathbb{P}^*(H_0) < \alpha\right) \sum_{s' \in \mathcal{S}} P^{M^*}_{\mu_k(s,j)}(s'|s) \left( V^{M_k}_{\mu_k^{\alpha*},j+1} - V^{M^*}_{\mu_k^{\alpha*},j+1} \right)(s') \Big| M_k, M^* \right]$$

$$+ \mathbb{E}\left[ I\left(\mathbb{P}^*(H_0) \geq \alpha\right) \sum_{s' \in \mathcal{S}} P^{M^*}_{\pi(s,j)}(s'|s) \left( V^{M_k}_{\mu_k^{\alpha*},j+1} - V^{M^*}_{\mu_k^{\alpha*},j+1} \right)(s') \Big| M_k, M^* \right]$$

$$- \mathbb{E}\left[ \left( V^{M_k}_{\mu_k^{\alpha*},j+1} - V^{M^*}_{\mu_k^{\alpha*},j+1} \right)(s_{j+1}) \Big| M_k, M^* \right]$$

$$= \left( I\left(\mathbb{P}^*(H_0) < \alpha\right) \sum_{s' \in \mathcal{S}} P^{M^*}_{\mu_k(s,j)}(s'|s) + I\left(\mathbb{P}^*(H_0) \geq \alpha\right) \sum_{s' \in \mathcal{S}} P^{M^*}_{\pi(s,j)}(s'|s) \right) \left( V^{M_k}_{\mu_k^{\alpha*},j+1} - V^{M^*}_{\mu_k^{\alpha*},j+1} \right)(s')$$

$$- \left( I\left(\mathbb{P}^*(H_0) < \alpha\right) \sum_{s' \in \mathcal{S}} P^{M^*}_{\mu_k(s,j)}(s'|s) + I\left(\mathbb{P}^*(H_0) \geq \alpha\right) \sum_{s' \in \mathcal{S}} P^{M^*}_{\pi(s,j)}(s'|s) \right) \left( V^{M_k}_{\mu_k^{\alpha*},j+1} - V^{M^*}_{\mu_k^{\alpha*},j+1} \right)(s')$$

$$= 0,$$

which follows by the expectation conditional on $M_k$, $M^*$ and definition of policy $\mu_k^{\alpha*}$.

Putting I) and II) together we get

$$\mathbb{E}\left[\left(V_{\mu_k^{\alpha*},1}^{M_k} - V_{\mu_k^{\alpha*},1}^{M^*}\right)(s_1)\,\middle|\,M^*, M_k\right] = \mathbb{E}\left[\sum_{j=1}^{\tau} \mathcal{W}_j + e_j\,\middle|\,M^*, M_k\right]$$

$$= \mathbb{E}\left[\sum_{i=1}^{\tau}\left(\mathcal{T}_{\mu_k^{\alpha*}(\cdot,j)}^{M_k} - \mathcal{T}_{\mu_k^{\alpha*}(\cdot,j)}^{M^*}\right) V_{\mu_k^{\alpha*},j+1}^{M_k}(s_j)\,\middle|\,M^*, M_k\right]$$

$\square$

*Proof of Lemma C.2.* First consider Azuma-Hoeffding's Inequality: Let $Z_1, Z_2, \ldots$ be a martingale sequence difference with $|Z_j| \leq c \,\forall j$. Then $\forall \epsilon > 0$ and $n \in \mathbb{N}$ $P\left[\sum_{i=1}^{n} Z_i > \epsilon\right] \leq \exp\left\{-\frac{\epsilon^2}{2nc^2}\right\}$. By definition the difference between the estimated transition and reward functions and their true respective functions are:

$$\hat{P}_a(s'|s) - P_a^M(s'|s) = \frac{\sum_{i,j\in\mathcal{I}}\left(I(s_{i,j+1} = s') - P_a^M(s'|s)\right) I(s_{ij} = s, a_{ij} = a)}{N_T(s,a)},$$

$$\hat{R}(s,a) - R^M(s,a) = \frac{\sum_{i,j\in\mathcal{I}}(r_{ij} - R^M(s,a)) I(s_{ij} = s, a_{ij} = a)}{N_T(s,a)},$$

now let $\tilde{\beta}_T(s,a) \equiv \sqrt{8ST\log(2TSA)}$, and consider the transition probability function, for a fixed state action pair $(s,a)$, let $\boldsymbol{\xi} = (\xi(s_1), \ldots, \xi(s_S)) \in \{-1, 1\}^S$, we have

$$\mathbb{P}\left(\sum_{s'\in\mathcal{S}}\left|\frac{\sum_{i,j\in\mathcal{I}}\left(I(s_{i,j+1} = s') - P_a^M(s'|s)\right) I(s_{ij} = s, a_{ij} = a)}{N_T(s,a)}\right| \geq \frac{\tilde{\beta}_T(s,a)}{N_T(s,a)}\right)$$

$$\leq \mathbb{P}\left(\max_{\boldsymbol{\xi}\in\{-1,1\}^s}\sum_{s'\in\mathcal{S}}\xi(s')\sum_{i,j\in\mathcal{I}}\left(I(s_{i,j+1} = s') - P_a^M(s'|s)\right) I(s_{ij} = s, a_{ij} = a) \geq \tilde{\beta}_T(s,a)\right)$$

$$\leq 2^S\mathbb{P}\left(\sum_{s'\in\mathcal{S}}\sum_{i,j\in\mathcal{I}}\xi(s')\left(I(s_{i,j+1} = s') - P_a^M(s'|s)\right) I(s_{ij} = s, a_{ij} = a) \geq \tilde{\beta}_T(s,a)\right)$$

where the first step follows from multiplying by $N_T(s,a)$, and eliminating the absolute value with $\boldsymbol{\xi}$, we use a union bound for the second step as there are $2^S$ possible $\boldsymbol{\xi}$ for a fixed $(s,a)$ pair. Next we use Azuma-Hoeffding's inequality to bound the $2^S$ probability terms, note that within the probability function we are summing over $T$ terms:

$$2^S\mathbb{P}\left(\sum_{s'\in\mathcal{S}}\sum_{i,j\in\mathcal{I}}\xi(s')\left(I(s_{i,j+1} = s') - P_a^M(s'|s)\right) I(s_{ij} = s, a_{ij} = a) \geq \tilde{\beta}_T(s,a)\right)$$

$$\leq 2^S\exp\left\{-\frac{8ST\log(2TSA)}{2\times 2^2 T}\right\}$$

$$\leq 2^S\exp\left\{\log((2TSA)^{-S})\right\} = 2^S\frac{1}{(2TSA)^S} < \frac{1}{TSA},$$

next we sum over all $(s,a)$ pairs and get

$$\mathbb{P}\left(\left\|\hat{P}_a(s'|s) - P_a^M(s'|s)\right\|_1 \geq \beta_T(s,a)\right)$$

$$\leq \sum_{s\in\mathcal{S},a\in\mathcal{A}}\mathbb{P}\left(\sum_{s'\in\mathcal{S}}\left|\frac{\sum_{i,j\in\mathcal{I}}\left(I(s_{i,j+1} = s') - P_a^M(s'|s)\right) I(s_{ij} = s, a_{ij} = a)}{N_T(s,a)}\right| \geq \frac{\tilde{\beta}_T(s,a)}{N_T(s,a)}\right)$$

$$\leq SA\frac{1}{TSA} = \frac{1}{T},$$

which follows from using a union bound again. Analogous we can show that $\mathbb{P}\left(\left|\hat{R}(s,a) - R^M(s,a)\right| \geq \beta_T(s,a)\right) \leq \frac{1}{T}$, thus

$$\mathbb{P}\left(M^* \notin \mathcal{M}_T\right), \mathbb{P}\left(M_T \notin \mathcal{M}_T\right) < \frac{1}{T}.$$

$\square$