[Reviews · NeurIPS 2020]

Review 1

Summary and Contributions: The paper proposes the hypothesis testing as a solution for combining the expert knowledge and the knowledge learned by RL in batch RL setting. The paper proposes a Bayesian approach to this problem in which the learned policy actions is only preferred to the expert actions when there exists a high confidence that it is better otherwise the null hypothesis (expert recommendation) is used. To choose between the expert policy and the learned policy, the algorithm samples from the posterior multiple times and compare the performance of the learned policy with the expert policy. If the learned policy is better than the expert on large majority of these samples then the algo chooses the learned policy otherwise it reverts to the expert policy. The paper provides some performance guarantees in terms of regret bounds and also evaluate the idea on 2 domains of riverswim (standard toy problem in RL) and a medical dataset on Spesis.

Strengths: I think the main idea of combining the expert and learned knowledge based on the confidence about the learned policy is quite intuitive and it can provide a good framework for safe and sample efficient RL in batch setting. Also the fact that the algorithmic idea is accompanied by theoretical results and experiments on the quite interesting and relevant domain of Spesis is very much appreciated.

Weaknesses: I think despite the good potentials for the idea of this paper, it suffers from the lack of clarity in its mathematical derivations and theoretical results to the extent that it makes the evaluation of the contributions quite difficult. There seems to be some errors in the derivations and definitions, and the notation are not defined clearly. More importantly I am not sure the way that the algo chooses between expert policy and the learned policy is the best way of doing it. Basically in the current version the algorithm counts the number of time that the expert policy and the learned optimal policy in agreement. If this number exceeds a certain threshold then it retain the expert decision (This is not clear from the way that the algorithm is described but when investigated more closely it seems this is what is happening). I am not sure this is the best strategy for combining the expert and the learned strategy. This is because when the probability of the optimality of expert action is low it can be due to the fact that all actions have low probability of being optimal due to uncertainty and not necessarily because the expert strategy is not as good as others. One may argue that this is exactly the situation that one wants to use the expert policy as opposed to high-variance learned policy.

Correctness: The definition of regret in line 184 seems not to be correct. With the current definition the bounds of Thm. 3.4 trivially holds as |V(s)|\leq \tau, which is of course not very interesting. I believe some notion of summation over t is missing in the definition of regret. Also \mu_k is used in line 164 before being defined, only later in Algorithm 1 table there is reference to its definition.

Clarity: The text is clear but the mathematical derivations and the theoretical results can be improved a lot in terms of clarity of notation and description of the idea behind mathematical derivations. The existing description of the algorithm looks very confusing to me. It seems to me that based on the current algo and definitions (and I can not say it for sure due to lack of clarity) that in line 162-163 the null hypothesis only can hold when \mu_k(s,a)=\pi(s,a), otherwise it can not hold due to the fact that \mu_k(s,a) is the maximizer of Q^(k)_\alpha. This basically means that the algorithm simply counts the number of times that the expert policy and the learned policy agree and use it as the measure to choose between the expert and the learned policy. If this is the case why it is not explicitly said so. Also no intuition is provided as why the dataset is split in partitions I_1 and I_2. It is just like it comes out of nowhere. Also in line 171-172 one of the I_1s I believe should be I_2.

Relation to Prior Work: I think the paper makes a good link between this work and prior work on Bayesian RL. But it can improve a lot in terms of covering the prior work on imitation learning and learning from expert which is quite relevant to this work.

Reproducibility: No

Additional Feedback: It seems to me that combining the strategies based on confidence intervals is a more natural way of combining the expert and learned policy: if the confidence intervals of expert and the learned optimal policy overlap significantly then use the expert otherwise use learned optimal policy. I wonder maybe this can be combined with the proposed algo to guarantee that in the case of high uncertainty the expert action is prioritized. ##### Post Rebuttal Authors have provided satisfactory response on the main concern with regard to the way Algo chooses between the expert and the learned policy and they have corrected some errors which led to it. However I think the presentation of the method and the Algo. should be improved in the final version to clarify this concern as the current presentation of method is quite confusing.


Review 2

Summary and Contributions: The paper proposes ESRL, a bayesian RL approach for offline RL. Specifically, ESRL utilizes value uncertainty obtained by sampling MDPs from posterior for hypothesis testing with a specified risk aversion threshold. Furthermore, the samples from the learned value posterior can be used to visualize the uncertainty which can be useful in high-stakes RL domains such as healthcare.

Strengths: > The paper is well-motivated and tackles the important challenge of providing uncertainty estimates for decision-making in offline RL contexts. > The use of hypothesis testing seems to be a novel incorporation in offline RL training. > The theoretical result regarding the regret bounds for ESRL indicates its sample efficiency when provided with a sampling mechanism for the MDP distribution (priors and likelihoods for rewards and dynamics). > The interpretability, based on value function posteriors, may be quite useful for healthcare domains as shown on the Sepsis medical dataset in Figure 4.

Weaknesses: > The empirically evaluation misses relevant baselines, making it quite hard to evaluate the usefulness of ESRL in comparison to prior approaches. 1. The main algorithm (Algo 1) incorporates the use of majority voting and hypothesis testing in addition to learning multiple Q-estimates based on K sampled MDPs. Furthermore, based on the figure captions, K seems to be large (250 for Riverswim, 500 for Sepsis) and it seems unfair to use a single DQN model. A *naive* baseline would be to use the ensemble of these K Q-estimates and simply use their mean for selecting actions: this *quantifies* the empirical benefit from hypothesis testing. 2. Related to ensembles and uncertainty estimation, ensembles has been studied before in the context of offline RL[1] on discrete Atari games and it was found that random ensembles (REM) significantly outperform DQN/naive ensembling in the offline setting. This should be discussed in the paper as well as empirically compared to as should be made as this is a simple way to incorporate value uncertainty in offline RL. 3. As mentioned in the paper, ESRL can deviate from the behavior policy when required or stick to it depending on the hypothesis testing. This makes me wonder how much better ESRL is able to perform than simply doing *behavior cloning* on the offline dataset. This is found to be quite effective on a lot of tasks in offline RL benchmarks such as RL Unplugged[2] and D4RL[3] and as well as on Atari when the data is collected from a single noisy behavior policy [4]. 4. As mentioned in the paper, a lot of prior approaches incorporate regularization towards the behavior policy for offline RL. This is akin to the risk aversion \alpha used in the paper. Unfortunately, comparison to such papers is not done by this paper as it is mentioned that these methods are for continuous spaces, however there are discrete versions of these algorithms exist too (which the authors seem to be unaware of) -- for example, see discrete BCQ[4, 2]. The paper mentions that " there are cases .. such regularization will lead to sub-optimal actions" (Line 87-88), however, it is unclear to me if that would happen in practice as the regularization penalty can be picked carefully similarly to the risk-aversion threshold. References: [1] Agarwal, Rishabh, et al. "An optimistic perspective on offline reinforcement learning". ICML (2020) [2] Gulcehre, Caglar, et al. "RL Unplugged: Benchmarks for Offline Reinforcement Learning." arXiv preprint arXiv:2006.13888 (2020). [3] Fu, Justin, et al. "D4rl: Datasets for deep data-driven reinforcement learning." arXiv preprint arXiv:2004.07219 (2020) [4] Fujimoto, Scott, et al. "Benchmarking batch deep reinforcement learning algorithms." arXiv preprint arXiv:1910.01708 (2019).

Correctness: Theoretically -- yes but empirical validation seems incomplete [see weakness section above].

Clarity: Yes.

Relation to Prior Work: Some of the relevant prior work is not discussed as well as skipped in the empirical evaluation (but easily fixable I suppose): Fujimoto, Scott, et al. "Benchmarking batch deep reinforcement learning algorithms." arXiv preprint arXiv:1910.01708 (2019). Agarwal, Rishabh, et al. "An optimistic perspective on offline reinforcement learning". ICML (2020) Some other recent (but less relevant) peer-reviewed offline RL work is not discussed in the paper and I'd recommend the authors to do so. This blog post may be a good starting point: https://danieltakeshi.github.io/2020/06/28/offline-rl/

Reproducibility: Yes

Additional Feedback: Post Rebuttal Response: The rebuttal has addressed most of my concerns about the empirical baselines and I'd like to recommend the paper for acceptance. I highly encourage the authors to include this new comparisons in the appendix and well as incorporate their response in the main paper. Minor: I was surprised that DQNE outperforms REM and I think it's likely if you aren't using separate ensemble networks in REM similar to that of DQNE. If not, please add some justification for this happens. ------------- I would be happy to raise my score if the authors can run the suggested baselines for a more thorough empirical assessment for ESRL or provide stronger justifications for why the proposed baselines are unwarranted. Since these baselines are quite easy to code (especially as the open-source code is available for most of them), hopefully it shouldn't be much hassle to report their performance. Additional questions: 1. It seems that for sampling MDPs from a distribution f(. | D_T), we need access to a prior, likelihoods for transitions and rewards. Can you please comment about the applicability of ESRL to domains without access to such models (e.g., real world robotics tasks or Atari)? 2. I was confused by the title "Expert-supervised RL" -- what/who is considered as the expert here and how is that providing supervision? Is it simply the offline dataset being considered as expert -- if so, it is confusing as experts are usually associated with demonstrations or imitation learning rather than reinforcement learning. [Minor] Typos: Line 177: "and I_1 for .." --> "and I_2 for .." Line 223: "state spaces" -- > "action spaces" (Atari also effectively has a continuous state space, but DQN still works there which is also a baseline you used)


Review 3

Summary and Contributions: This paper proposes a Bayesian-based approach for off policy RL: expert-supervised RL (ESRL). Based on expert-collected data, the method uses hypothesis testing at each state to decide whether switching to the optimal action (derived from the posterior) or just following expert's action. The hypothesis testing is performed by an independent set of samples drawn from the posterior. Instead of just following the posterior model, the method only utilizes the model-based approach if there is enough evidence that the action will be better than expert's demonstration, which makes it will be useful in some high risk domains, such as dynamic treatment in health care, autonomous driving.

Strengths: This paper tackles the problem of conservative off-policy learning using hypothesis testing. The idea is simple and easy to understand, the intuition is well-explained through the health-care example. Theoretically, the paper examines the consistency/unbiasedness of the estimated Q-function (using posterior sampling) and the bayesian type regret bound. Empirically, the performance of ESRL is examined in riverswim and sepsis datasets, with robust learning performance under different stochasticity level of the behaviour policy.

Weaknesses: - I am concerned about several aspects of the problem set-up. It seems we need to have the prior distribution f coincides with the distribution of true MDP M*, and it seems that most of the theoretical properties: unbiasedness and regret bound relies on this assumption. Actually this assumption looks kind of strong too me, in practice it will be impossible for us to know the distribution of the true MDP, otherwise the problem will be easily solved (if I understood it correctly). How the things break up without this assumption? Also, another minor issue is the behaviour policy is epsilon-greedy, but I feel the result should also goes through if we relax it to general stochastic policy? Meanwhile, It will be great to have more details on how to choose prior distributions? - For the sepsis experiments, Figure 4 illustrates the posterior distribution for several Q functions. A more direct measure may be the overall learning performance, like reward? How's the reward for different baselines? Also for the OPPE, ESRL seems like model-based evaluation, how it compares with other model-based estimators, it seems IS/WIS are purely sample based? Also, it would be great to have experiments in more complex domains.

Correctness: The claims and method seems correct given the assumption. The empirical methodology seems good, however not comprehensive enough.

Clarity: I feel the paper messed with some typos that kind of affecting the reading. Also,

Relation to Prior Work: The author did a great job on literature review, however I feel this work also has a strong connection which imitation learning, it will be great to see the comparison in this line of research.

Reproducibility: Yes

Additional Feedback: Some minor issues: - conditional expectation in Lemma 3.1 is wired. - line 177 uses the same notation for different sets. - Algorithm 1, when calculating the null probability, does the letter k with different font mean something differently?


Review 4

Summary and Contributions: An interesting ensemble algorithm for offline RL where “posteriors are used to test the hypothesis that the optimal action is indeed better than the one from the behavior policy” with ambitious theoretical analysis which includes a regret bound formulated in Theorem 3.4.

Strengths: 1. Simple and interesting offline RL algorithm. 2. A very good example from medicine motivating the design of the algorithm - it is also matched by selection of the Sepsa environment in the experimental section.

Weaknesses: Bigger problems: 1. environments used for experiments look somewhat naive (Riverswim) or contrived (Sepsa). For the Sepsa environment it is also quite hard to understand what is a strong baseline. 2. No comparison with other offline RL algorithms (BCQ, AWR, BEAR, …) 3. DQN can be a bit weak model-free baseline. Minor problems: 1. The impact and meaning of the regret bound is not discussed in the paper. 2. The algorithm relies on a manual selection of a hyperparameter alpha which decides about the risk aversion of the agent. 3. The description of the related work is scattered through the paper - most relevant papers such as [14,15] are in place, but I would be glad to see a broader discussion how this algorithm compares to other offline RL algorithms. 4. Experiments in my opinion are hard to interpret - perhaps authors can improve readability and captions of figures.

Correctness: Both mathematical and empirical parts look correct to me.

Clarity: This is a very well written paper.

Relation to Prior Work: The discussion is sufficient, but see remarks above.

Reproducibility: Yes

Additional Feedback: Post-rebuttal feedback: thank you for adding new experiments and environments. I think that the paper deserves publication though I would like to see the algorithm benchmarked on more difficult environments.

[Author Response · NeurIPS 2020]

We thank the reviewers for their constructive criticism, we address the common and particular points below:

**Baselines:** Following the suggestions, we have added Taxi-v2 and FrozenLake-v0 environments as they have discrete
state and actions spaces. Results will be shown in the main manuscript appendix. Additionally, we added the following
offline methods for comparison: a) an ensemble of $K$ DQN models (DQNE); b) behavior cloning (BC), c) Batch
Constrained deep Q-learning (BCQ), and d) Random Ensemble Mixture (REM). We also include discussion on these
and imitation learning methods to the updated manuscript. Fig. 1 a) Shows mean reward on Riverswim for offline
training on $T$=200 episodes, we vary stochasticity in behavior policy $\pi$. Fig 1 b) shows posterior distribution for $\hat{V}(\hat{\mu})$
on Sepsis data. ESRL performs well across quality in $\pi$ in a) and b). BC does well in Riverswim but not on Sepsis, on
the contrary BCQ, and REM do better on Sepsis: a more complex and higher variance setting with more training data.

**Reviewer 1:** Thank you for pointing out the typos and hard to follow notation, we have corrected and simplified these
to make derivations easier to follow. **1)** Regarding the p-value in Algorithm 1, we apologize for an unfortunate typo
($\mu_{\mathbb{K}}$ should be $\hat{\mu}_k$) and the confusion this caused. To clarify, we first estimate $\hat{\mu}(s,t)$ with sample $\mathcal{I}_1$ and use that to
compare $Q_t^{(k)}(s, \hat{\mu}(s,t))$ and $Q_t^{(k)}(s, \pi(s,t))$ where $k \in \mathcal{I}_2$. It is correct that we are simply counting, but the subtlety
is that the cross sample strategy allows for $Q_t^{(k)}(s, \hat{\mu}(s,t)) < Q_t^{(k)}(s, \pi(s,t))$. **2)** With respect to the combination of
the expert and learned policy, the proposed strategy of using confidence intervals (CI) is appealing. We argue that both
this strategy, and our method which relies on the p-value are equivalently optimal. As any $(1-\alpha)\%$ CI can be inverted
to obtain an associated $\alpha$-level hypothesis test, it can be shown that measuring CI overlap is equivalent to a hypothesis
test. **3)** You mention an important example: when the expert action is optimal but presented in a high variance setting.
The information theoretical lower bound tells us that in this high variance settings, the high noise-to-signal ratio will
lead towards failure of rejecting $H_0$ due to lack of statistical power. ESRL would reject $H_0$ and choose the expert's
action, same decision would come from a CI comparison, which in this case would be highly overlapping.

**Reviewer 2:** Thank you very much for the thorough suggestion on baselines, please see the results above for the
methods you proposed. We cite the relevant papers in the main text. **1)** You raised an important question on access to
$f(\cdot|\mathbf{D}_T)$ for sampling MDPs in real-world problems and Atari. As data increases, dependency of results on the prior
decreases, so we can use any *working model* to approximate the MDP. Several models are computationally simple to
sample from, and can be used for learning. For example, we use the Dirichlet/multinomial, and normal-gamma/normal
conjugates which work well for all settings in the paper. Chosen priors should be flexible enough to capture the dynamics
and easy to sample from efficiently. **2)** Regarding the Expert supervision in the title, we allude to the healthcare setting
where physicians are experts, but contrary to imitation learning, their actions might be far from optimal.

**Reviewer 3:** Thank you for the careful look at the theoretical results. **1)** We certainly agree that the prior specification
involves a strong assumption. However this is usual the case with all model based RL, and in particular the Bayesian
aspect of our method alleviates this issue as the posterior distribution is robust to prior model specification. This follows
as the posterior distribution of samples $M_k$ will concentrate around $M^*$ as sample size increases. The error induced
from using $M_k$ instead of $M^*$ vanishes, as shown in the proof of Theorem 3.4. Please see response to Reviewer 2.1 for
discussion on choosing a model. **2)** As you mention, ESRL works for stochastic behavior policies as well, thank you
for pointing this out. **3)** The $\mu_{\mathbb{K}}$ in Algorithm 1 should be $\hat{\mu}$. **4)** Thank you for suggesting OPPE benchmarks, we added
two non-parametric approximate-model comparisons to the manuscript, plots are not shown here due to lack of space.
**5)** For Sepsis we visualize the $\hat{V}(\hat{\mu})$ in figure 4(c) in the manuscript, and have changed the caption to make this clearer.

**Reviewer 4:** Thank you for the suggestions on adding more comparison methods and environments, please see Baseline
section for results, we cited the suggested papers in the main manuscript. **1)** Since our main motivation is the healthcare
domain, we chose the Sepsis data set as it is standard in a lot of prior work on RL for healthcare (please see [1],[2],[26]
in the manuscript). The common baseline is the physician's policy $\pi$. We chose Riverswim as it is has discrete state and
action spaces, and requires deep exploration to reach high rewards. We added openAI's Taxi-v2 and FrozenLake-v0 to
the paper. **2)** We added more discussion on Theorem 3.4, mainly we highlight how it shows ESRL is sample efficient,
and flexible to risk aversion and stochasticity level of $\pi$. **3)** We believe that the risk aversion parameter $\alpha$ is actually
a benefit; it allows the method to be tailored to different scientific and business applications where one might have
different tolerance towards risk in search for more reward. **4)** We have improved readability and captions of the figures.

[Meta-Review · NeurIPS 2020]

This paper proposes an interesting way to use hypothesis testing as a solution to use expert knowledge for offline RL. The proposed approach is exciting and good enough to be published at NeurIPS. The experimental results are interesting, as well. However, the authors should address the concerns on the presentation and theoretical results raised by Reviewer 1 in the camera-ready version of the paper. At the very least, discussing it is the limitation of the approach in the paper's conclusion.